**The composition and distribution of semi-labile dissolved organic matter across the South**
**West Pacific**
Christos Panagiotopoulos[1]*, Mireille Pujo Pay[2], Mar Benavides[1], France Van Wambeke[1], and
Richard Sempéré[1]
[1] *Aix-Marseille Université, Université de Toulon, CNRS, IRD, Mediterranean Institute of*
*Oceanography (MIO), UM 110, 13288, Marseille, France*
[2] *Laboratoire d'Océanographie Microbienne (LOMIC), Observatoire Océanologique, Sorbonne*
*Universités, UPMC Univ. Paris 06, CNRS, 66650 Banyuls/Mer, France*
*Corresponding author e-mail: christos.panagiotopoulos@mio.osupytheas.fr
Submitted to Biogeosciences (OUTPACE special issue)
16 October 2018

**Abstract**

The distribution and dynamics of dissolved organic carbon (DOC) and dissolved combined neutral sugars (DCNS) were studied across an increasing oligotrophic gradient (-18 to -22°N latitude) in the Tropical South Pacific Ocean, spanning from the Melanesian Archipelago (MA) area to the western part of the South Pacific gyre (WGY), in austral summer, as a part of the OUTPACE project. Our results showed DOC and DCNS concentrations exhibited no statistical differences between the MA and WGY areas (0-200 m: 47-81 μMC for DOC and 0.2-4.2 μMC for DCNS). However, due to a deepening of the euphotic zone, a deeper penetration of DOC was noticeable at 150 m depth at the WGY area. This finding was also observed with regard to the excess-DOC ($DOC_{EX}$), which was determined as the difference between surface and deep-sea DOC values. Euphotic zone integrated stocks of both DOC and $DOC_{EX}$ were higher in the WGY than the MA area. Considering $DOC_{EX}$ as representative of the semi-labile DOC ($DOC_{SL}$), its residence time was calculated as the $DOC_{SL}$ to bacterial carbon demand (BCD) ratio. This residence time was $176 \pm 43$ days (n = 3) in the WGY area, about three times longer than in the MA area ($T_r = 51 \pm 13$ days (n = 8)), suggesting an accumulation of the semi-labile dissolved organic matter (DOM) in the surface waters of WGY. Average epipelagic (0-200 m) DCNS yields (DCNS x $DOC^{-1}$), based on volumetric data, were roughly similar in both areas, accounting for ~2.8% of DOC. DCNS exhibited a longer residence time in WGY ($T_r = 91 \pm 41$ days, n = 3) than in MA ($T_r = 31 \pm 10$ days, n=8) further suggesting that this DCNS pool persists longer in the surface waters of the WGY. The accumulation of $DOC_{EX}$ in the surface waters of WGY is probably due to the very slow bacterial degradation due to nutrient/energy limitation of heterotrophic prokaryotes indicating that biologically produced DOC can be stored in the euphotic layer of the South Pacific gyre for a long period.

**1. Introduction**

Gyres are oceanic deserts similar to those found in continental landscapes spanning an area of several thousands of Km and are characterized by low nutrient content and limited productivity (Raimbault et al., 2008; D'Hondt et al., 2009; Bender et al., 2016; de Verneil et al., 2017, 2018). Moreover, gyres are now considered as the world's plastic dumps (Law et al., 2010; Eriksen et al., 2013; Cozar et al., 2014), whereas their study may us help to understand future climate changes (Di Lorenzo et al., 2008; Zhang et al., 2014) and marine ecosystem functioning (Sibert et al., 2016; Browing et al., 2017). Among the five well-known oceanic gyres the South Pacific gyre, although the world's largest, has been less extensively studied mainly due to its remoteness from the main landmasses. Nonetheless, earlier studies indicated that Western Tropical South Pacific (WTSP) is a hot spot of $N_2$ fixation (Bonnet et al., 2013; Bonnet et al., 2017; Caffin et al., 2018) and recent studies have shown that there is a gradient of increasing oligotrophy from WTSP to the western part of the Pacific gyre (Moutin et al., 2018). The ultra-oligotrophic regime is reached in the center of the gyre, and then it decreases within the eastern part of the gyre toward the Chilean coast (Claustre et al., 2008) with high residual phosphate concentrations in the center of the gyre (Moutin et al., 2008).

Recent studies indicated that efficient DOC export in the subtropical gyres is related with the inhibition of DOC utilization under low-nutrient conditions (Letscher et al., 2015; Roshan and DeVries, 2017). Similar patterns have been observed for the oligotrophic Mediterranean Sea (Guyennon et al., 2015). However, little information exists regarding dissolved organic matter (DOM) dynamics in the south Pacific gyre particularly for its semi-labile component (accumulation, export, fate), which is mainly represented by carbohydrates (Sempéré et al., 2008; Goldberg et al., 2011; Carlson and Hansell, 2015).

Among the three well-identified chemical families (amino acids, lipids and carbohydrates) in

seawater, carbohydrates are the major components of organic matter in surface and deep waters
accounting for 5-10% and < 5% of dissolved organic carbon (DOC), respectively as shown by
liquid chromatography (Benner, 2002; Panagiotopoulos and Sempéré, 2005 and references therein).
The carbohydrate pool of DOC consists of free monosaccharides, oligosaccharides and
polysaccharides. Major polysaccharides are constituted by dissolved combined neutral sugars
(DCNS), which are generally measured as their monosaccharide constituents (sum of fucose,
rhamnose, arabinose, galactose, glucose, mannose and xylose) after acid hydrolysis (McCarthy et
al., 1996; Aluwihare et al., 1997; Skoog and Benner, 1997; Kirchman et al., 2001; Panagiotopoulos
and Sempéré, 2005). Other minor carbohydrate constituents of DOC include the amino sugars
(glucosamine, galactosamine and muramic acid; Benner and Kaiser, 2003), uronic acids (glucuronic
and galacturonic acids; Hung et al., 2003; Engel and Handel, 2011), methylated and dimethylated
sugars (Panagiotopoulos et al., 2013), heptoses (Panagiotopoulos et al., 2013) and sugar alcohols
(Van Pinxteren et al., 2012).
Free monosaccharide concentrations range from 10 to100 nM; they account < 10% of total
dissolved neutral sugars (TDNS), and experiments have shown that they are rapidly utilized
(minutes to hours) by bacterioplankton and as such they are considered as labile organic matter
(Rich et al., 1996; Skoog et al., 1999; Kirchman et al., 2001). Polysaccharide or dissolved combined
neutral sugars (DCNS) concentrations range from 200-800 nM; they account for 80-95% of TDNS
and experiments have shown that they disappear within time scales of days to months and, as such,
they are considered as labile and semi-labile organic matter (Aluwihare and Repeta, 1999; Carlson
and Hansell, 2015 and references therein). Other studies have shown that this labile and/or semi-
labile organic matter accumulates in the surface ocean and may potentially be exported to depth
contributing to the ocean carbon pump (Goldberg et al., 2010; Carlson and Hansell, 2015).
In the frame of the OUTPACE project we studied DOM dynamics in terms of DOC and DCNS
composition and tried to evaluate their residence time. The results are presented and discussed
along with heterotrophic prokaryotic production in order to better understand the bacterial cycling
of DOM  in the region.

**2. Materials and Methods**
**2.1 Sampling**
Sampling took place along a 5500 Km transect spanning from New Caledonia to French
Polynesia in the WTSP aboard the R/V *L'Atalante* during the Oligotrophy to Ultraoligotrophy
Pacific Experiment (OUTPACE) cruise (19 February-5 April, 2015). Samples were taken from 18
different stations comprising three long duration stations (LDA, LDB, and LDC; about 7-8 days)
and 15 short duration (SD1-15) stations (~8 h). Biogeochemical and physical characteristics of
these sites are described in detail elsewhere (Moutin et al., 2017). Briefly, the cruise took place
between 18-20°S covering two contrasted trophic regimes with increasing oligotrophy from west to
east (Fig. 1).
Discrete seawater samples were collected from 12 L Niskin bottles equipped with Viton O-rings
and silicon tubes to avoid chemical contamination. For DOC and DCNS analyses, samples were
filtered through two pre-combusted (450°C for 24 h) GF/F filters using a custom-made all-
glass/Teflon filtration syringe system. Samples for DOC (SD: 1-15 including LD: A, B ,C) were
collected into precombusted glass ampoules (450°C, 6h) that were sealed after acidification with
$H_3PO_4$ (85%) and stored in the dark at 4°C. Samples for DCNS (SD 1, 3-7, 9, 11, 13-15 including
LD: C) were collected in 40-mL Falcon vials (previously cleaned with 10% of HCl and Milli-Q
water) and frozen at -20°C until analysis.

**3. Chemical and microbiological analyses**
**3.1. Dissolved organic carbon (DOC) determination**
DOC was measured by high temperature combustion on a Shimadzu TOC-L analyzer (Cauwet,
1999). Typical analytical precision was ± 0.1-0.5 µM C (SD) for multiple injections (3-4) of
replicate samples. Consensus reference materials were injected every 12 to 17 samples to ensure
stable operating conditions and were in the range 42-45 µM (lot # 07-14;
(http://yyy.rsmas.miami.edu/groups/biogeochem/Table1.html).

**3.2. Dissolved combined neutral sugars (DCNS) determination**
*3.2.1. Carbohydrate extraction and isolation*

Seawater samples were desalted using dialysis tubes with a molecular weight cut-off of 100-500

Da (Spectra/Por® Biotech cellulose ester) according to the protocol of Panagiotopoulos et al.
(2014). Briefly, the dialysis tube was filled with 8 mL of the sample and the dialysis was conducted
into a 1 L beaker filled with Milli-Q water at 4°C in the dark. Dialysis was achieved after 4-5 h
(salinity dropped from 35 to 1-2 g $L^{-1}$). Samples were transferred into 40 mL plastic vials (Falcon;
previously cleaned with 10% HCl and Milli-Q water), frozen at -30 °C, and freeze dried. The
obtained powder was hydrolyzed with 1M HCl for 20 h at 100°C and the samples were again freeze
dried to remove the HCl acid (Murrell and Hollibaugh, 2000; Engel and Handel, 2011). The dried
samples were diluted in 4 mL of Milli-Q water, filtered through quartz wool, and pipetted into
scintillation vials for liquid chromatographic analysis. The vials were kept at 4°C until the time of
analysis (this never exceeded 24 h). The recovery yields of the whole procedure (dialysis and
hydrolysis) were estimated using standard polysaccharides (laminarin, and chondroitine sulfate) and
ranged from 82 to 86% (n=3). Finally, it is important to note that the current desalination procedure
does not allow the determination of the dissolved free neutral sugars (i.e., sugar monomers present
in samples with MW ~ 180 Da) because these compounds are lost/poorly recovered during the
dialysis step (Panagiotopoulos et al., 2014).

*3.2.2. Liquid Chromatography*
Carbohydrate concentrations in samples were measured by liquid chromatography according to
Mopper et al. (1992) modified by Panagiotopoulos et al. (2001, 2014). Briefly, neutral
monosaccharides were separated on an-anion exchange column (Carbopac PA-1, Thermo) by
isocratic elution (mobile phase 19 mM NaOH) and were detected by an electrochemical detector set
in the pulsed amperometric mode (Panagiotopoulos et al., 2014). The flow rate and the column
temperature were set at 0.7 mL min$^{-1}$ and 17°C, respectively. Data acquisition and processing were
performed using the Dionex software Chromeleon. Repeated injections (n = 6) of a dissolved
sample resulted in a CV of 12-15% for the peak area, for all carbohydrates. Adonitol was used as an
internal standard and was recovered at a percentage of 80-95%; however, we have chosen not to
correct our original data.

**3.3. Bacterial production**

Heterotrophic prokaryotic production (here abbreviated classically as "bacterial" production
or BP) was determined onboard with the $^{3}$H-leucine incorporation technique to measure protein
synthesis (Smith and Azam, 1992). Additional details are given in Van Wambeke et al. (2018).
Briefly, 1.5 mL samples were incubated in the dark for 1-2 h after addition of $^{3}$H leucine, at a final
concentration of 20 nM, with standard deviation of the triplicate measurements being on average
9%. Isotopic dilution was checked and was close to 1 (Van Wambeke et al, 2018), and we therefore
applied a conversion factor of 1.5 Kg C mol leucine$^{-1}$ to convert leucine incorporation to carbon
equivalents (Kirchman, 1993). BP was corrected for leucine assimilation by *Prochlorococcus*
(Duhamel et al., 2018) as described in Van Wambeke et al. (2018). To estimate bacterial carbon
demand (BCD) which is used to calculate semi-labile DOC residence time, we used a bacterial
growth efficiency (BGE) of 8% as determined experimentally using dilution experiments during the
OUTPACE cruise (Van Wambeke et al., 2018). BCD was calculated by dividing BP values at each
station by BGE.  Euphotic zone integrals were then computed from volumetric rates.

**4. Results**

4.1 General observations

The OUTPACE cruise was conducted under strong stratification conditions (Moutin et al.,

2018) during the austral summer encompassing a longitudinal gradient starting at the oligotrophic
Melanesian Archipelago (MA waters; stations SD1-SD12 including LDA and LDB stations) and
ending in the ultra-oligotrophic western part of the South Pacific gyre (WGY waters; stations SD13-
SD15 including LDC station; Fig. 1). Additional information on the hydrological conditions of the
study area (*i.e* temperature, salinity) including water masses characteristics is provided elsewhere
(de Verneil et al., 2018; Moutin et al., 2018). Mixed layer depth ranged from 11 to 34 m with higher
values recorded in the WGY (Moutin et al., 2018). The depth of the deep chlorophyll maximum
ranged from 69 to 119 m and from 122 to 155 m for the MA and WGY areas, respectively. Two
different trends can be noticed in a first approach:

a. Most of the biogeochemical parameters examined in the OUTPACE cruise (chlorophyll $\alpha$

concentrations, primary production, BP, BCD, $N_2$ fixation rates, and nutrient concentrations)
showed significantly higher values in the MA area than in the WGY area (Moutin et al., 2018; Van
Wambeke et al., 2018; Benavides et al., 2018; Caffin et al., 2018). These differences were also
reflected by the distribution of the diazotrophic communities detected in both areas further
highlighting the different dynamics across the oligotrophic gradient (Stenegren et al., 2018; Moutin
et al., 2017, 2018). The net heterotrophic/autotrophic status of the MA and WGY areas has been
discussed in previous investigations by comparing BCD and gross primary production (GPP) (Fig.
2). By using propagation of errors, Van Wambeke et al. (2018) concluded that GPP minus BCD
could not be considered different from zero at most of the stations investigated (11 out of 17)
showing a metabolic balance. For the other stations, net heterotrophy was shown at stations SD 4, 5,
6 and LDB, and net autotrophy at station SD9 (Van Wambeke et al, 2018).

b. The bulk of DOM as shown by DOC analysis did not follow the above biogeochemical

pattern and showed little variability on DOC absolute concentrations although a deeper penetration
of DOM was noticeable at 150 m depth in the WGY area (Fig. 3a; Table 1). As such, epipelagic (0-
200 m) DOC concentrations throughout the OUTPACE cruise ranged from 47 to 81 µM C (mean ±
sd: 67 ± 10 µM; n = 136) except at LDB (~85 µM C) which is probably related to a decaying
phytoplankton bloom (de Verneuil et al., 2018; Van Wambeke et al., 2018). Mesopelagic (200-1000
m) DOC values varied between 36 to 53 µM C (mean ± sd: 46 ± 4 µM; n = 67) (Fig. 4a; Table 1)
and are in agreement with previous studies in the South Pacific Ocean (Doval and Hansell, 2000;
Hansell et al., 2009; Raimbault et al. 2008).

DCNS concentrations closely followed DOC trends and fluctuated between 0.2-4.2 µM C

(mean ± sd: 1.9 ± 0.8 µM; n = 132) in the epipelagic zone (Fig. 3b; Table 1). These values are in
good agreement to those previously reported for the central and/or the eastern part of the South
Pacific gyre (1.1-3.0 µM C; Sempéré et al., 2008) that were recorded under strong stratification
conditions during austral summer (Claustre et al., 2008). Compared with other oceanic provinces
our epipelagic DCNS concentrations fall within the same range of those reported in the BATS
station in the Sargasso Sea (1.0-2.7 µM C) also monitored under stratification conditions (Goldberg
et al., 2010). Mesopelagic DCNS concentrations ranged from 0.3 to 2.4 µMC (average ± sd: 1.2 ±
0.6 µM; n = 68) (Fig. 4b; Table 1) and concur with previously reported literature values at the
ALOHA station (0.2-0.8 µMC; Kaiser and Benner, 2009) or in the Equatorial Pacific (0.8-1 µMC;
Skoog and Benner, 1997).

4.2 DCNS yields and composition

223 The contribution of DCNS-C to the DOC pool is referred to here as DCNS yields and is

224 presented as a percentage of DOC (*i.e* DCNS-C x DOC$^{-1}$ %). Epipelagic (0-200 m) average DCNS

225 yields, based on volumetric data, were similar between the WGY (range 0.3-5.1%; average ± sd: 2.8

226 ± 1.3%; n = 41) and MA (range 0.8-7.0%; average ± sd: 2.8 ± 1.0%; n = 91) areas whereas deeper

227 than 200 m they were 2.4 ± 1.0% (n = 23) and 2.7 ± 1.3% (n = 43)  for the WGY and MA,

228 respectively (Table 1). These values are in good agreement to those reported for the eastern part of

229 the gyre (Sempéré et al., 2008) and concur well with the range of values (2-7%) recorded in the

230 Equatorial Pacific (Rich et al., 1996; Skoog and Benner, 1997).

231 The molecular composition of carbohydrates revealed that glucose was the major

232 monosaccharide at all depths in both the MA and WGY areas accounting on average for 53 ± 18%

233 (n = 132) of the DCNS in epipelagic waters and 64 ± 21% (n = 68) in mesopelagic waters (Table 1).

234 Epipelagic glucose concentrations (DCGlc-C) averaged 1.0 ± 0.6; n = 132 in both areas (Fig. 3c,

235 Table 1), however, a significantly higher mol% contribution of glucose was recorded in the WGY

236 than the MA especially at depths > 200 m (Fig. 5). Glucose was followed by xylose (9-12%),

237 galactose (4-9%) and mannose (5-8%) whereas the other monosaccharides accounted for < 6% of

238 DCNS (Fig. 5). The same suite of monosaccharides was also reported by Sempéré et al. (2008)

239 although the latter author also found that arabinose was among the major monosaccharides. Finally,

240 it is worth noting that the relative abundance of glucose increased with depth and sometimes

241 accounted 100% of the DCNS (Table 1, Fig. 5).


243 4.3 DOC and DCNS integrated stocks


245 DOC stocks (euphotic zone integrated) were calculated at the same stations where carbohydrate

246 (DCNS) data were available and were compared between the MA (stations: SD 1, 3, 4, 5, 6, 7, 9,

247 11) and WGY (SD13-SD15; LDC) stations (Fig. 6). DOC stock values in the euphotic were 9111 ±

1159 (n = 8) and 13266 ± 821 (n = 4) mmol C m$^{-2}$ for the MA and WGY areas, respectively. Excess
DOC stock (DOC$_{EX}$) was calculated by subtracting an average deep DOC value from the bulk
surface DOC pool. This DOC value was 40 µMC and was estimated averaging all DOC values
below 1000 m depth from all stations (39.6 ± 1.4 µMC, n = 36). DOC$_{EX}$ stock values averaged
3717 ± 528 (n = 8) and 5265± 301 (n = 4) mmol C m$^{-2}$ accounting about 40% of DOC in both areas.
DCNS represented 6.7 and 7.1% of DOC$_{ex}$ in the MA and WGY sites, respectively, further
suggesting that only a small percentage of DOC$_{EX}$ can be attributed to DCNS (polysaccharides).

**5. Discussion**

5.1 DOC and DCNS stocks in relation with biological activity

Euphotic zone integrated stocks of DOC, DOC$_{EX}$ and DCNS were respectively 46, 42 and 52%
higher in the WGY than in the MA (Fig. 6), as opposed to BCD and GPP (Fig. 2). This is a
consequence of the deepening of the euphotic zone, because the variability of the volumetric stocks
was high, and not statistically different in the euphotic zone between MA and WGY areas. As
indicated above DOC$_{EX}$ is calculated as the difference between the bulk surface DOC and deep
DOC the latter assumed to be refractory. Thus, DOC$_{EX}$ is often described as "semi-labile" DOC or
DOC$_{SL}$ with a turnover on time scales of weeks to months (Carlson and Hansell, 2015). DCNS
belong to this semi-labile category of DOC (Biersmith and Benner, 1998; Aluwihare et Repeta,
1999; Benner, 2002), and the results of this study showed that DCNS represented a low proportion
(~7%) of DOC$_{EX.}$ Because the conditions of the HPLC technique employed in this study does not
allow identification and quantification of all the carbohydrate components of DOC (methylated
sugars, uronic acids, amino sugars etc) it is possible that the contribution of polysaccharides to the
DOC$_{EX}$ is underestimated. Previous investigations on amino sugars and methylated sugars indicated
that these monosaccharides account for < 3% of the carbohydrate pool (Benner and Kaiser;
Panagiotopoulos et al., 2013) while uronic acids may account for as much as 40% of the
carbohydrate pool (Engel et al., 2012) indicating that the latter compounds should at least be
considered in future DOM lability studies.

Other semi-labile compounds that potentially may contribute to the $DOC_{EX}$ pool are proteins

and lipids. Unfortunately, proteins (combined amino acids) were not measured in this study.
Nonetheless, previous investigations indicated that total dissolved amino acids represent 0.7-1.1%
of DOC in the upper mesopelagic zone of the north Pacific (Kaiser and Benner, 2012) further
suggesting a relatively small contribution of amino acids to the $DOC_{EX}$. During the OUTPACE
cruise, assimilation rates of $^{3}H$- leucine using concentration kinetics were determined (Duhamel et
al., 2018) and, based on the Wright and Hobbie (1966) protocol, the ambient concentration of
leucine was determined. The results showed a lower ambient leucine concentration at the LDC
(0.56 nM) than at the LDA (1.80 nM) stations (Duhamel et al., 2018).

This result may suggest that single amino acid and perhaps proteins concentrations are very low

at the LDC station, reflecting the ultra- oligotrophic regime of the WGY. On the other hand, DOM
exhibited only slightly different C/N ratios between MA (C/N = 13) and WGY (C/N =14), which
does not suggest differences in DON dynamics in relation with organic matter lability (data from
integrated values of 0-70 m; Moutin et al., 2018). Clearly further investigations are warranted on
combined and free amino acids distribution in relation with $N_2$ fixation.

The high stock of $DOC_{EX}$ measured in WGY was also characterized by an elevated residence

time ($T_{r\,SL}$) calculated as the ratio of $DOC_{EX}$ / BCD. This ratio is calculated based on the
assumption that $DOC_{EX}$ is representative of the $DOC_{SL}$ and the latter  pool turnover is at the scale
of seasonal mixing (i.e weeks to months) whereas the BP,  as determined with leucine technique on
short incubation times (1-2 hours), tracks only the ultra-labile to labile organic matter consumption
and not $DOC_{SL}$ utilization. Biodegradation experiments (3 experiments, duration 10 days each)
performed during the OUTPACE cruise showed that the labile DOC represented only 2.5 to 5% of
the DOC pool (Van Wambeke et al., 2018), confirming that the residence time calculated from
$DOC_{EX}$ / BCD overestimates the residence time of ultra-labile DOC. The bacterial production and
BGEs associated with the use of semi-labile DOC is currently not technically measurable due to
long-term confinement artifacts.  Our results showed that $T_{r\,SL}$ in the WGY was in the order of 176
$\pm$ 43 days (n = 3), i.e. about three times higher than in the MA region ($T_{r\,SL}$= 51 $\pm$ 13 days (n = 8))
indicating an accumulation of the semi-labile DOM in the surface waters of WGY (Fig. 7). As
suggested by previous studies the accumulation of DOC in the surface waters of oligotrophic
regimes may be related in biotic and/or abiotic factors.

Nutrient limitation can prevent DOC assimilation by heterotrophic bacteria and as such

sources and sinks are uncoupled, allow accumulation (Thingstad et al., 1997; Jiao et al., 2010; Shen
et al., 2016). Biodegradation experiments (Van Wambeke et al., 2018) focusing on the
determination of the BGE and the degradation of the labile DOC pool (turning over 10 days)
revealed a less biodegradable DOM fraction and lower degradation rates at the LDC (2.4% labile
DOC; 0.012 $d^{-1}$) than the LDA site (5.3% labile DOC; 0.039 $d^{-1}$). Other experiments, focusing on
the factors limiting BP by testing the effect of different nutrient additions, showed that over a short-
time period, BP is initially limited by the availability of labile carbon in the WGY (as tracked with
glucose addition, Van Wambeke et al., 2018). This limitation on BP by labile carbon/energy was
also the case at the center of the South Pacific gyre (Van Wambeke et al., 2008), while N limitation
(as tracked by addition of ammonium+nitrate) was more pronounced in the MA area.
Although extensive photodegradation may transform recalcitrant organic matter into labile, the
low content in chromophoric DOM recorded in the surface waters of WGY ($\alpha$CDOM(350) = 0.010-
0.015 $m^{-1}$, 0-50 m; Dupouy et al. unpublished results from the OUTPACE cruise) points toward an
already photobleached and thus photodegraded organic material (Tedetti et al., 2007; Carlson and
Hansel, 2015). Notably, the 10% irradiance depths for solar radiations (Z 10%) clearly showed a
higher penetration of UV-R and PAR radiations in the WGY area than in MA area (Dupouy et al.,

2018). These results are in agreement with previous investigations reporting intense solar radiation in the South Pacific gyre highlighting an strong decrease of chromophoric dissolved organic matter (CDOM) in the gyre (Tedetti et al., 2007). Less energy available for heterotrophic prokaryotes should prevent them from degrading such recalcitrant, photo-degraded organic matter.

The computation of the carbon, nitrogen, and phosphorus budgets in the upper 0-70 m layer by Moutin et al. (2018) suggested that at 70 m the environmental conditions remained seasonally unchanged during the OUTPACE cruise, forming an average wintertime depth of the mixed layer. These authors calculated seasonal (from winter to austral summer) net DOM and POM accumulation on the basis of such assumptions, and found a dominance of DOC accumulation in the MA area (391 to 445 mmol $m^{-2}$ over 8 months). This DOC accumulation in the MA area was 3.8 to 8.1 times higher than that of POC accumulation during the same time period. On the other hand, only DOC accumulated at WGY, although the amount was two times lower in magnitude than in the MA (391- 445 vs 220 mmol $m^{-2}$). The accumulation of DOC and $DOC_{EX}$ (Fig. 6) in the WGY may have important implications with regard to the sequestration of this organic material in the mesopelagic layers. DOC appears to be the major form of export of carbon in the WGY area and this result agrees with the general feature observed in oligotrophic regimes (Roshan and Devries, 2017).

5.2 DCNS dynamics across the South West Pacific

Previous investigations have employed the DCNS yields along with mol% of glucose to assess the diagenetically "freshness" of organic matter (Skoog and Benner, 1997; Benner, 2002; Goldberg et al. 2010). In general freshly produced DOM has DCNS yields >10% and mol% glucose between 28-71% (Biersmith and Benner, 1998; Hama and Yanagi, 2001). Elevated mol% glucose (> 25%) does not necessarily mirror fresh material because such values have also been reported for deep

DOM and low molecular weight DOM that are considered as a diagenetically altered material
(Skoog et al., 1997).
Our results showed that epipelagic DCNS yields were about similar (~2.8%) in both WGY and
MA areas (Table 1) further indicating a similar contribution of DCNS to the DOC pool despite the
major differences observed for the other biochemical parameters (e.g. deepening of the nitraclines
and deep chlorophyll maximum etc) between MA and WGY. As expected, DCNS yields decreased
by depth but were always comparable between WGY and MA areas (Table 1). By analogy to the
$DOC_{SL}$, we tried to estimate a DNCS residence time assuming that (a) the ectoenzymatic hydrolysis
is a rate-limiting step for bacterial production, ii) the mean contribution of polysaccharides
hydrolysis to bacterial production is 11%, based on Pointek et al. (2011), and iii) this 11%
correction factor can be propagated to BCD. On the basis of these assumptions, we estimated a
DCNS residence time as DCNS/(11% x BCD). The results showed that DCNS exhibited a higher
residence time in the WGY ($T_{r\,DCNS-C}$= 91 ± 41 days, n = 3) than the MA area ($T_{r\,DCNS-C}$ = 31 ± 10
days, n = 8) which clearly shows that the DCNS pool persist longer in the surface waters of the
WGY (Fig. 7). Moreover, because carbohydrates do not absorb light these polysaccharides (DCNS)
do not seem to be impacted by the high photochemistry in WGY and potentially may be exported in
the Ocean interior during a non-stratification period (e.g. winter  time) considering their high
residence time at the WGY area. In addition, their slow utilization could also be related to energy
limitation by heterotrophic prokaryotes in the WGY area.
Glucose accounted for ~50% of DCNS in the MA surface waters which most likely reflects the
high abundance of *Trichodesmium* species in that area (Dupouy et al., 2018; Rousset et al., 2018). A
roughly similar percentage of glucose was also recorded in surface WGY waters (Fig. 5a) which is
probably due to the low utilization of semi-labile organic matter in the form of exopolysaccharides.
These exopolysaccharides are probably hydrolyzed by bacteria, but not taken up due to limited
nutrient availability. At 200 m depth, glucose accounted for 75% and 50% of DCNS in the WGY

and MA areas, respectively (200 m depth), and this percentage increased considerably with depth in both areas (76% for MA and 96% for WGY at 2000 m depth) indicating a preferential removal of the other carbohydrates relative to glucose (Fig. 5b; Fig. 5c). The low DCNS yields (~1%) at 2000 m depth along with the high % mol abundance of glucose clearly suggests the presence of diagenetically altered DOM and is consistent with previous investigations (Skoog and Benner, 1997; Goldberg et al. 2010; Golberg et al., 2011).

**6. Conclusions**

This study showed a rather uniform distribution of DOC and DCNS concentrations in surface waters across an increasing oligotrophic gradient in the South West Pacific Ocean during the OUTPACE cruise. Nevertheless, our results showed that DOC and $DOC_{EX}$ stocks were by ~40% in WGY than the MA area, accompanied with higher residence times in the WGY area suggesting an accumulation of semi-labile material in the euphotic zone of WGY. Although DCNS accounted a small fraction of $DOC_{SL}$ (~7%) our results showed that DCNS or polysaccharides also exhibited a higher residence time ($T_{r\ DCNS-C}$) in the WGY than in the MA area indicating that DCNS persist longer in the WGY. This $T_{r\ DCNS-C}$ is calculated on the basis of many assumptions on DNCS hydrolysis rates that were not practically determined, showing the need to estimate such fluxes in order to better estimate the dynamics of carbohydrates. Glucose was the major monosaccharide in both areas (51 - 55%) and its relative abundance increased with depth along with a decrease of the DCNS yields indicating a preferential removal of the other carbohydrates relative to glucose. Clearly further investigations are warranted to better characterize the semi-labile DOC pool in terms of combined and free amino acids distribution in relation with $N_2$ fixation.

**Acknowledgements**

This is a contribution of the OUTPACE (Oligotrophy from Ultra-oligoTrophy PACific Experiment) project lead by T. Moutin and S. Bonnet and funded by the French national research agency (ANR-14-CE01-0007-01), the LEFE-CyBER program (CNRS-INSU), the GOPS program (IRD) and CNES (BC T23, ZBC 4500048836). The OUTPACE cruise (http://dx.doi.org/10.17600/15000900) was managed by the MIO from Marseille (France). The authors thank the crew of the R/V L'Atalante for outstanding shipboard operation. G. Rougier and M. Picheral are thanked for their efficient help in CTD rosette management and data processing. C. Schmechtig is acknowledged for the LEFE CYBER database management. We also thank A. Lozingot for administrative aid for the OUTPACE project. The authors also acknowledge Prof. R. Benner and one anonymous reviewer for valuable comments and fruitful discussions. M.B. was funded by the People Programme (Marie Skłodowska-Curie Actions) of the European Union's Seventh Framework Programme (FP7/2007-2013) under REA grant agreement number 625185. C.P. received support from the PACA region (MANDARINE project, grant number 2008-10372) and Aix Marseille University (ORANGE project, FI-2011).

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

**Figure and Table captions:**

Figure 1: Sampling stations during the OUTPACE cruise. The white line shows the vessel
course (data from the hull-mounted ADCP positioning system). Stations and their respective
names (SD1-SD15 including LDA, LDB and LDC) are depicted in grey. Figure courtesy of T.
Wagener.

Figure 2: Integrated stocks of bacterial carbon demand (BCD) and gross primary production (GPP)
(mmol C m$^{-2}$ d$^{-1}$) over the euphotic zone. Data from Van Wambeke et al. (2018). Error bars
correspond to standard deviation of the different stations. * BCD and GPP were statistically
different between MA and WGY areas (Man-Whitney test, $p<0.05$).

Figure 3: Distribution of A: dissolved organic carbon (DOC); B: dissolved combined neutral sugars
(DCNS); and C: dissolved combined glucose (DCGlc) in the upper surface layer (0-200 m) of the
study area. DCNS and DCGlc concentration is given in carbon equivalents in order to have the
same unit as DOC. Long duration stations (LDA, LDB and LDC) are also indicated in each graph.
White and red circles indicate the mixed layer depth and deep chlorophyll maximum, respectively
for each station.

Figure 4: Depth profiles of A: DOC; B: DCNS; and C: DCGlc in the 0-2000 m layer of the study
area.

Figure 5: Average Mol percentage (mol %) of dissolved monosaccharides at A: surface; B: 200 m;
and C: 2000 m depth for MA and WGY areas. Monosaccharides abbreviations: Fuc.: Fucose;
Rha.:Rhamnose; Ara.: Arabinose; GlcN.: Glucosamine; Gal.: Galactose; Glc.: Glucose; Man.:
Mannose and Xyl.: Xylose.

Figure 6: Integrated carbon stocks (mmol C m$^{-2}$) over the euphotic zone carbon in terms of DOC,
$DOC_{EX}$ and DCNS-C. * DOC and $DOC_{SL}$ were statistically different between MA and WGY areas
(Man-Whitney test, $p<0.05$).

Figure 7: Residence time (days) of semi labile DOC ($T_{r\ SL}$) and DCNS-C ($T_{r\ DCNS-C}$) for MA and
WGY areas. * $T_{r\ SL}$ and $T_{r\ DCNS-C}$ were statistically different between MA and WGY areas (Man-
Whitney test, $p<0.05$).
Table 1: Range and mean values (0-200 m and 200-1000 m) of DOC (µMC), DCNS-C (µMC),
DCGlc-C (µMC), DCNS-C/DOC (%) and DCGlc-C/DCNS-C (%) recorded during the OUTPACE
cruise. MA comprises the SD2-SD12 stations and WGY comprises the LDC and SD13-SD15.
Means of MA and WGY were not statistically different for any of the parameters presented (Man-
Whitney test, p > 0.05).


Table 1: Range and mean values (0-200 m and 200-1000 m) of DOC, DCNS-C, DCGlc-C, DCNS-C/DOC and DCGlc-C/DCNS-C recorded during the OUTPACE cruise. MA comprises the SD2-SD12 stations and WGY comprises the LDC and SD13-SD15. Means of MA and WGY were not statistically different for any of the parameters presented (Man-Whitney test, p >0.05).

| | All data | | | | MA | | | | WGY | | | |
|---|---|---|---|---|---|---|---|---|---|---|---|---|
| | Range | mean±sd (n) | Range | mean±sd (n) | Range | mean±sd (n) | Range | mean±sd (n) | Range | mean±sd (n) | Range | mean±sd (n) |
| DOC (µM) | 47-81 | 67±10 (136) | 36-53 | 46±4 (67) | 51-79 | 66±9 (94) | 39-52 | 46±3 (43) | 47-81 | 68±10 (42) | 36-53 | 46±4 (24) |
| Depth (m) | 0-200 | | 200-1000 | | 0-200 | | 200-1000 | | 0-200 | | 200-1000 | |
| | | | | | | | | | | | | |
| DCNS-C (µM) | 0.2-4.2 | 1.9±0.8 (132) | 0.3-2.4 | 1.2 ±0.6 (68) | 0.6-4.2 | 1.8±0.7 (91) | 0.3-2.4 | 1.2±0.6 (45) | 0.2-3.8 | 1.9±1.0 (41) | 0.3-2.0 | 1.0±0.4 (23) |
| Depth (m) | 0-200 | | 200-1000 | | 0-200 | | 200-1000 | | 0-200 | | 200-1000 | |
| | | | | | | | | | | | | |
| DCGlc-C (µM) | 0.2-3.0 | 1.0±0.6 (132) | 0.2-1.6 | 0.7±0.3 (68) | 0.3-3.0 | 1.0±0.6 (91) | 0.2-1.6 | 0.7±0.4 (45) | 0.2-2.7 | 1.1±0.7 (41) | 0.3-1.4 | 0.7±0.3 (23) |
| Depth (m) | 0-200 | | 200-1000 | | 0-200 | | 200-1000 | | 0-200 | | 200-1000 | |
| | | | | | | | | | | | | |
| DCNS-C/DOC (%) | 0.3-7.0 | 2.8±1.1 (132) | 0.56-5.4 | 2.6±1.2 (66) | 0.8-7.0 | 2.8±1.0 (91) | 0.6-5.4 | 2.7±1.3 (43) | 0.3-5.1 | 2.8±1.3 (41) | 0.6-4.7 | 2.4±1.0 (23) |
| Depth (m) | 0-200 | | 200-1000 | | 0-200 | | 200-1000 | | 0-200 | | 200-1000 | |
| | | | | | | | | | | | | |
| DCGlc-C/DCNS-C (%) | 19-100 | 53±18 (132) | 35-100 | 64±21 (68) | 28-100 | 54±17 (91) | 36-100 | 63±22 (45) | 19-100 | 58±20 (41) | 35-100 | 66±20 (23) |
| Depth (m) | 0-200 | | 200-1000 | | 0-200 | | 200-1000 | | 0-200 | | 200-1000 | |

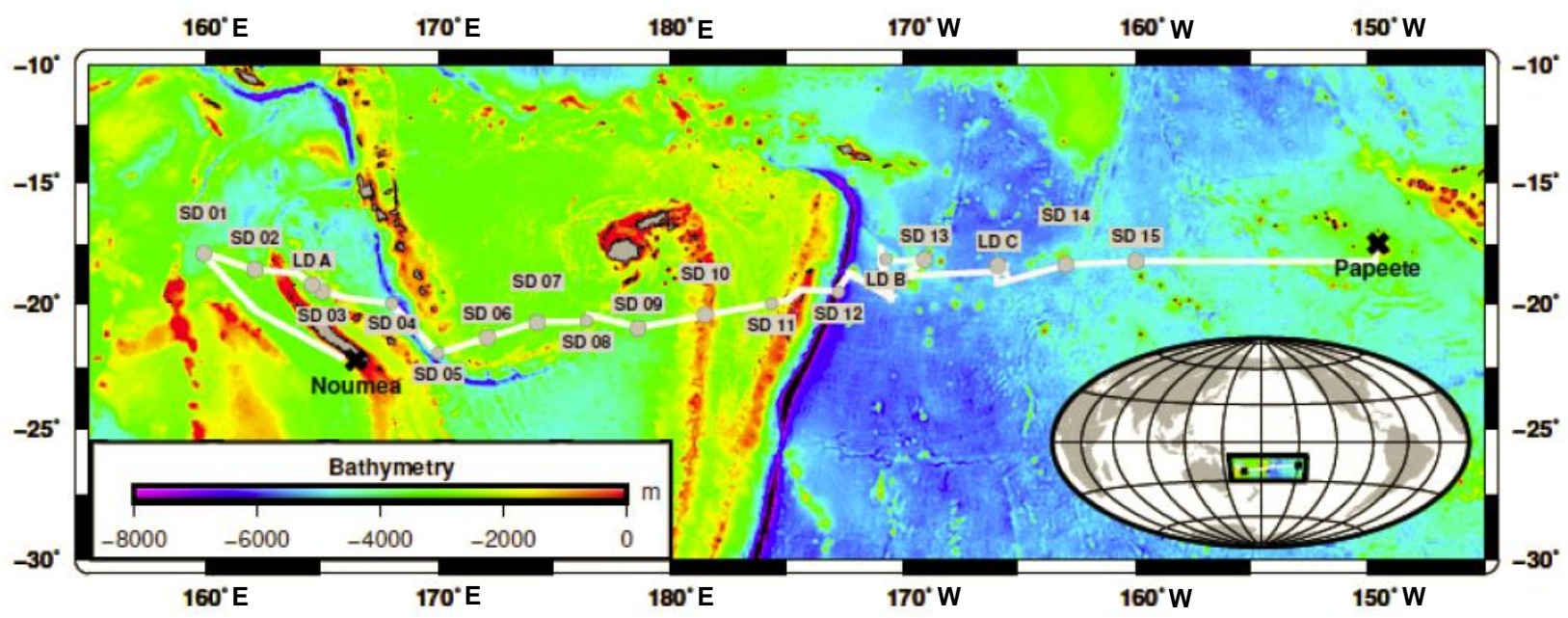

**Figure 1**

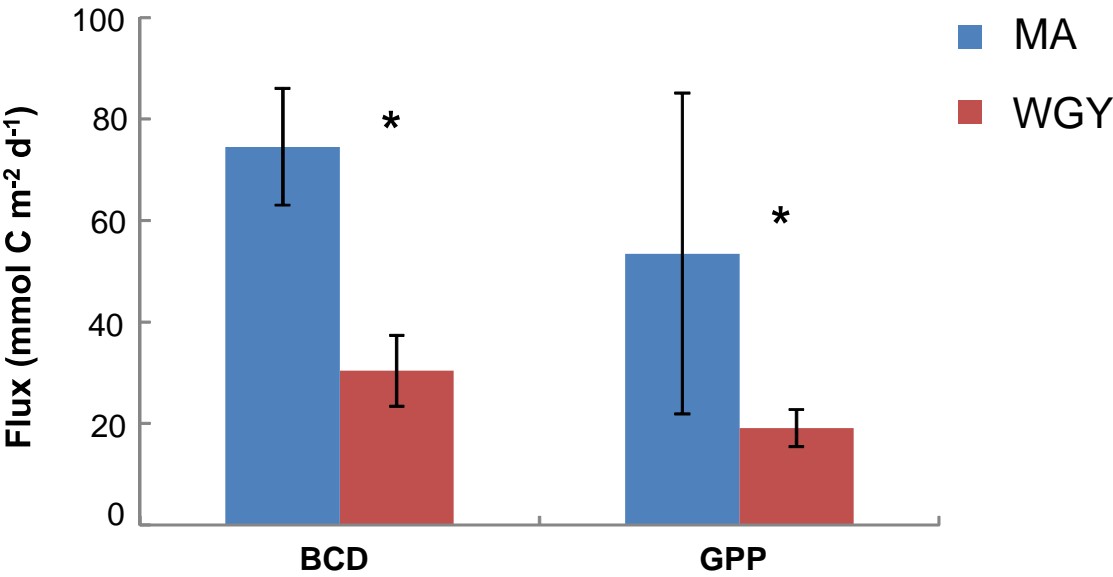

**Figure 2**

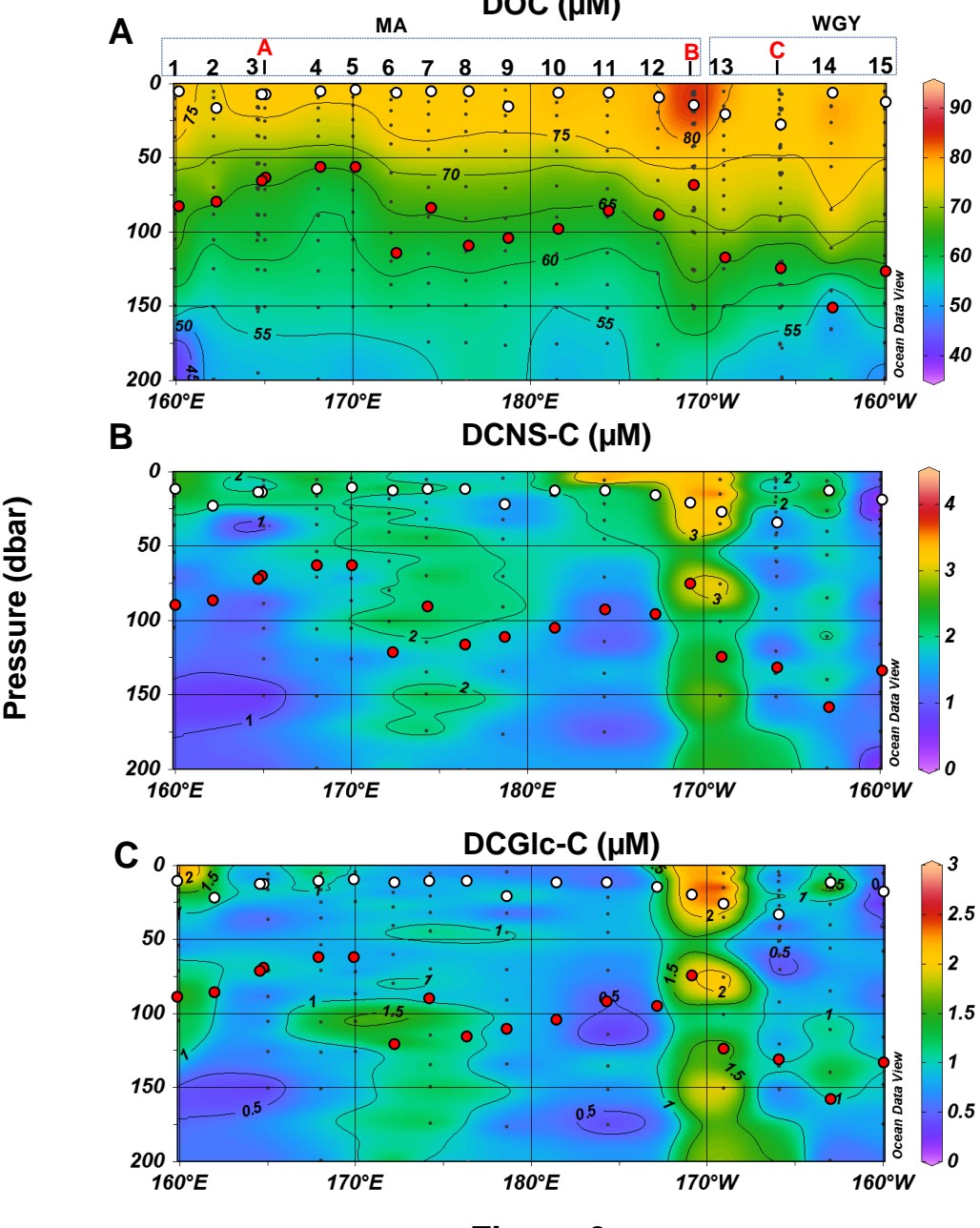

**Figure 3**

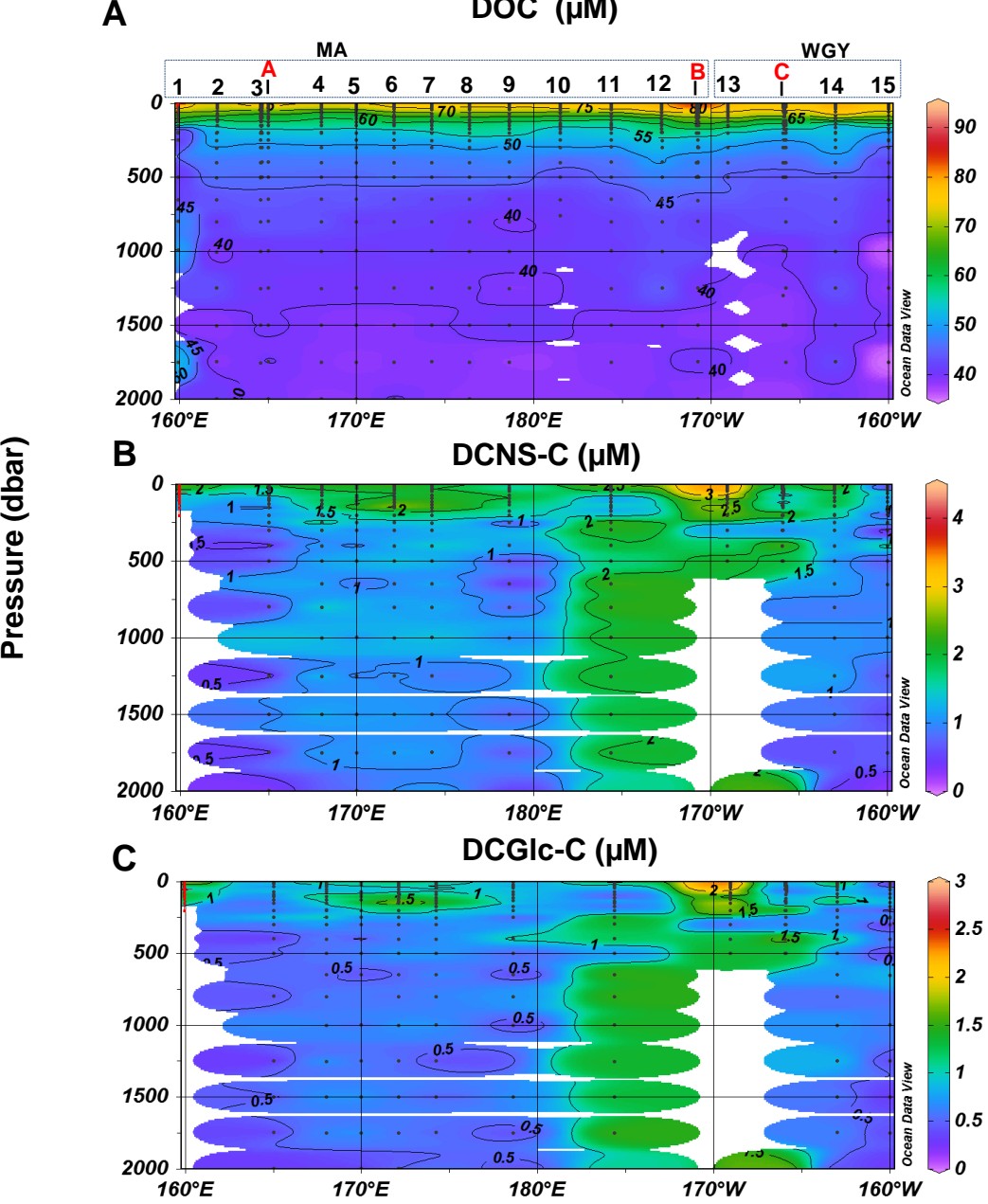

**Figure 4**

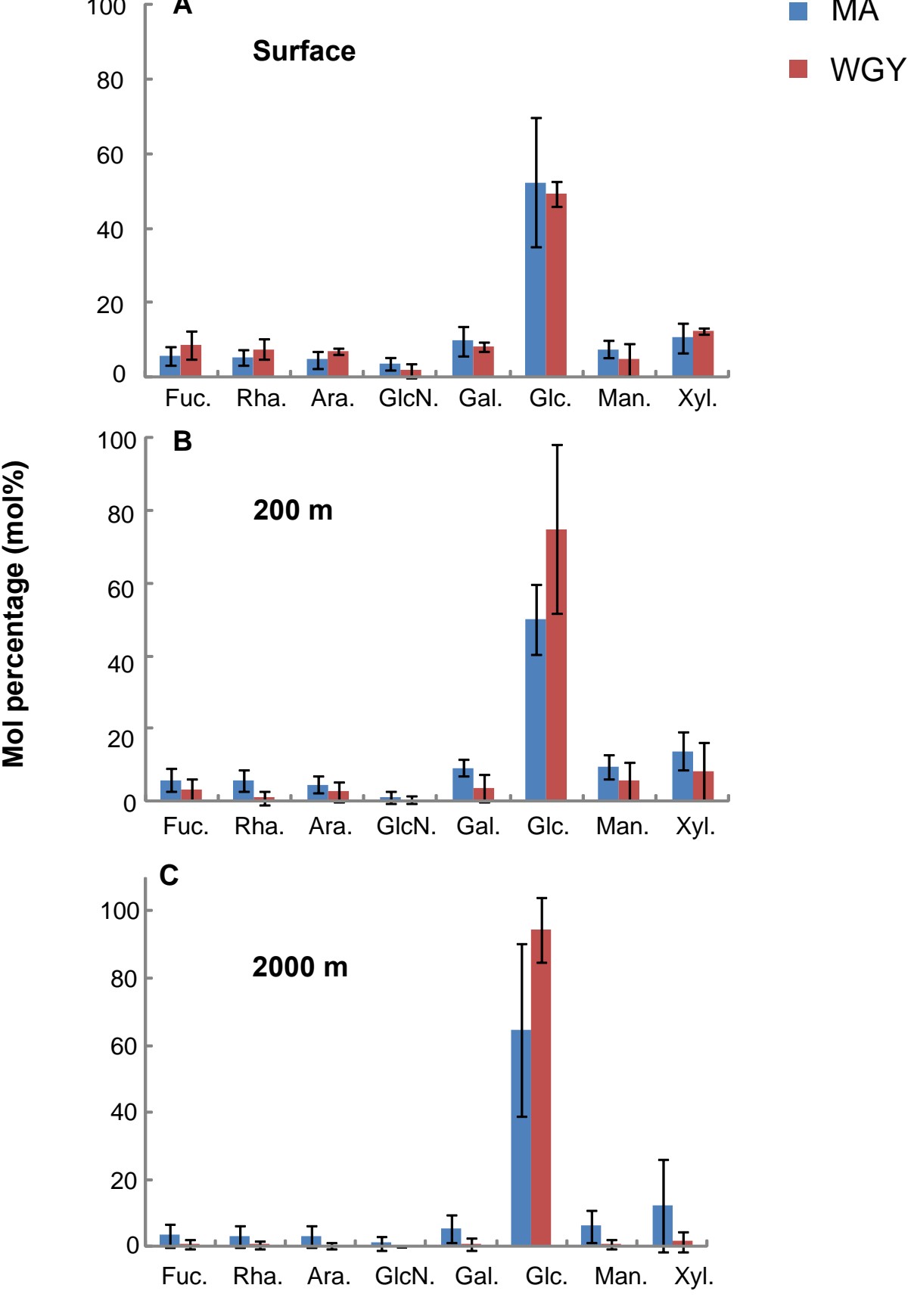

**Figure 5**

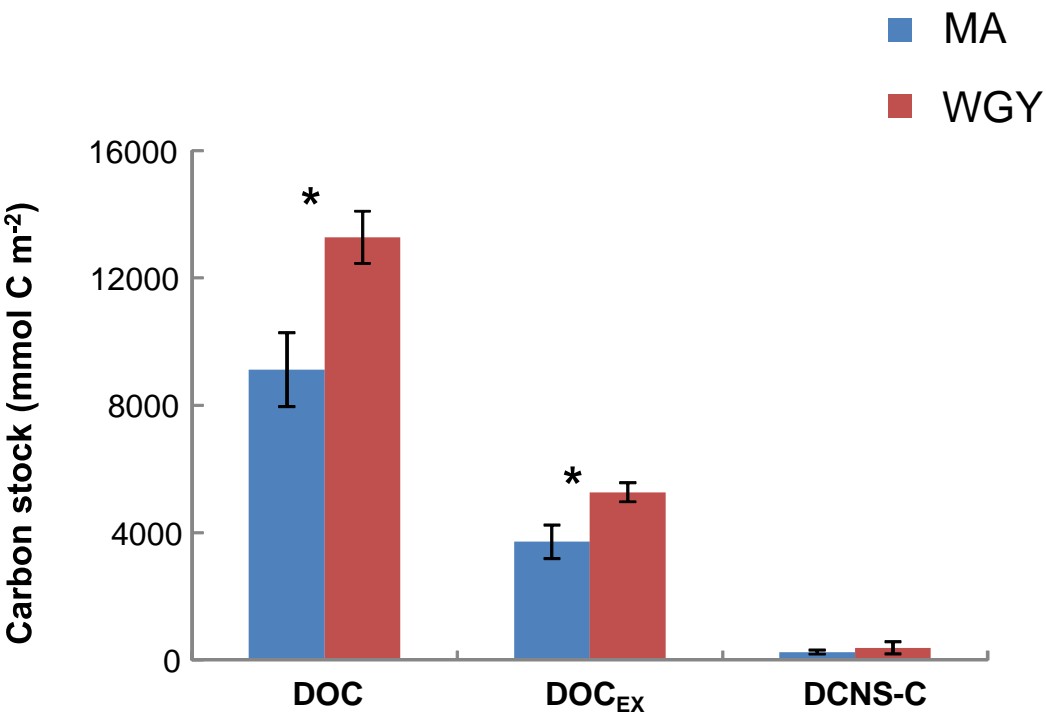

**Figure 6**

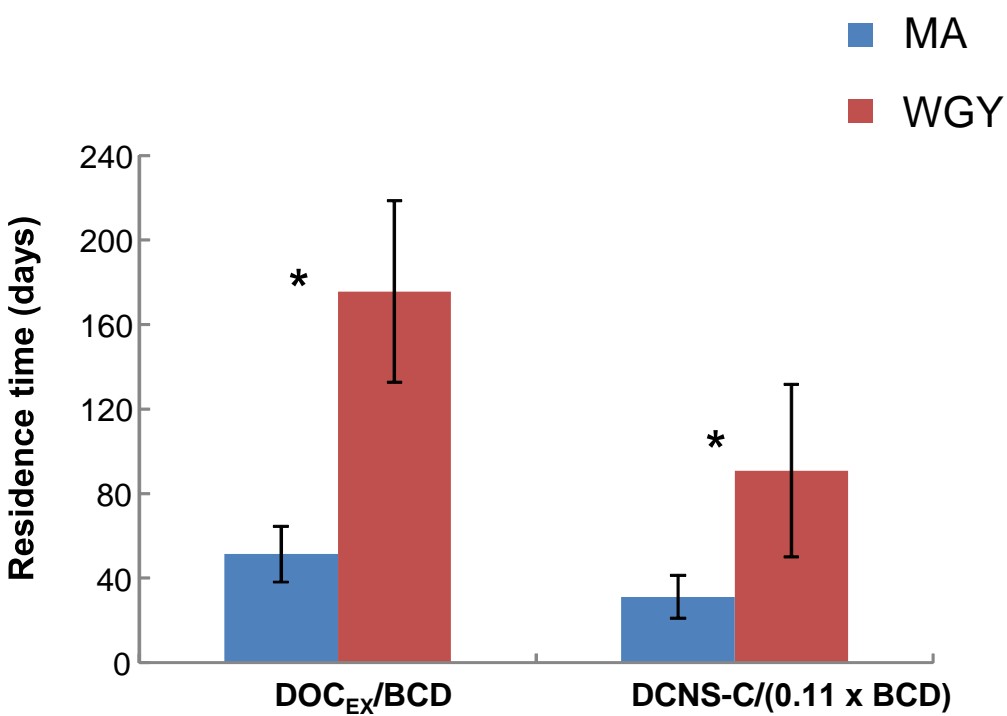

**Figure 7**

**The composition and distribution of semi-labile dissolved organic matter across the South**
**West Pacific**
Christos Panagiotopoulos[1]*, Mireille Pujo Pay[2], Mar Benavides[1], France Van Wambeke[1], and
Richard Sempéré[1]
[1] *Aix-Marseille Université, Université de Toulon, CNRS, IRD, Mediterranean Institute of*
*Oceanography (MIO), UM 110, 13288, Marseille, France*
[2] *Laboratoire d'Océanographie Microbienne (LOMIC), Observatoire Océanologique, Sorbonne*
*Universités, UPMC Univ. Paris 06, CNRS, 66650 Banyuls/Mer, France*
*Corresponding author e-mail: christos.panagiotopoulos@mio.osupytheas.fr
Submitted to Biogeosciences (OUTPACE special issue)
5 October 2018

**Abstract**

The distribution and dynamics of dissolved organic carbon (DOC) and dissolved combined neutral sugars (DCNS) were studied across an increasing oligotrophic gradient (-18 to -22°N latitude) in the Tropical South Pacific Ocean, spanning from the Melanesian Archipelago (MA) area to the western part of the South Pacific gyre (WGY), in austral summer, as a part of the OUTPACE project. Our results showed DOC and DCNS concentrations exhibited no statistical differences between the MA and WGY areas (0-200 m: 47-81 µMC for DOC and 0.2-4.2 µMC for DCNS). However, due to a deepening of the euphotic zone, a deeper penetration of DOC was noticeable at 150 m depth at the WGY area. This finding was also observed with regard to the excess-DOC ($DOC_{EX}$), which was determined as the difference between surface and deep-sea DOC values. Euphotic zone integrated stocks of both DOC and $DOC_{EX}$ were higher in the WGY than the MA area. Considering $DOC_{EX}$ as representative of the semi-labile DOC ($DOC_{SL}$), its residence time was calculated as the $DOC_{SL}$ to bacterial carbon demand (BCD) ratio. This residence time was $176 \pm 43$ days (n = 3) in the WGY area, about three times longer than in the MA area ($T_r = 51 \pm 13$ days (n = 8)), suggesting an accumulation of the semi-labile dissolved organic matter (DOM) in the surface waters of WGY. Average epipelagic (0-200 m) DCNS yields (DCNS x $DOC^{-1}$), based on volumetric data, were roughly similar in both areas, accounting for ~2.8% of DOC. DCNS exhibited a longer residence time in WGY ($T_r = 91 \pm 41$ days, n = 3) than in MA ($T_r = 31 \pm 10$days, n=8) further suggesting that this DCNS pool persists longer in the surface waters of the WGY. The accumulation of $DOC_{EX}$ in the surface waters of WGY is probably due to the very slow bacterial degradation due to nutrient/energy limitation of heterotrophic prokaryotes indicating that biologically produced DOC can be stored in the euphotic layer of the South Pacific gyre for a long period.

**1. Introduction**

Gyres are oceanic deserts similar to those found in continental landscapes spanning an area of several thousands of Km and are characterized by low nutrient content and limited productivity (Raimbault et al., 2008; D'Hondt et al., 2009; Bender et al., 2016; de Verneil et al., 2017, 2018). Moreover, gyres are now considered as the world's plastic dumps (Law et al., 2010; Eriksen et al., 2013; Cozar et al., 2014), whereas their study may us help to understand future climate changes (Di Lorenzo et al., 2008; Zhang et al., 2014) and marine ecosystem functioning (Sibert et al., 2016; Browing et al., 2017). Among the five well-known oceanic gyres the South Pacific gyre, although the world's largest, has been less extensively studied mainly due to its remoteness from the main landmasses. Nonetheless, earlier studies indicated that Western Tropical South Pacific (WTSP) is a hot spot of $N_2$ fixation (Bonnet et al., 2013; Bonnet et al., 2017; Caffin et al., 2018) and recent studies have shown that there is a gradient of increasing oligotrophy from WTSP to the western part of the Pacific gyre (Moutin et al., 2018). The ultra-oligotrophic regime is reached in the center of the gyre, and then it decreases within the eastern part of the gyre toward the Chilean coast (Claustre et al., 2008) with high residual phosphate concentrations in the center of the gyre (Moutin et al., 2008).

Recent studies indicated that efficient DOC export in the subtropical gyres is related with the inhibition of DOC utilization under low-nutrient conditions (Letscher et al., 2015; Roshan and DeVries, 2017). Similar patterns have been observed for the oligotrophic Mediterranean Sea (Guyennon et al., 2015). However, little information exists regarding dissolved organic matter (DOM) dynamics in the south Pacific gyre particularly for its semi-labile component (accumulation, export, fate), which is mainly represented by carbohydrates (Sempéré et al., 2008; Goldberg et al., 2011; Carlson and Hansell, 2015).

Among the three well-identified chemical families (amino acids, lipids and carbohydrates) in

seawater, carbohydrates are the major components of organic matter in surface and deep waters
accounting for 5-10% and < 5% of dissolved organic carbon (DOC), respectively as shown by
liquid chromatography (Benner, 2002; Panagiotopoulos and Sempéré, 2005 and references therein).
The carbohydrate pool of DOC consists of free monosaccharides, oligosaccharides and
polysaccharides. Major polysaccharides are constituted by dissolved combined neutral sugars
(DCNS), which are generally measured as their monosaccharide constituents (sum of fucose,
rhamnose, arabinose, galactose, glucose, mannose and xylose) after acid hydrolysis (McCarthy et
al., 1996; Aluwihare et al., 1997; Skoog and Benner, 1997; Kirchman et al., 2001; Panagiotopoulos
and Sempéré, 2005). Other minor carbohydrate constituents of DOC include the amino sugars
(glucosamine, galactosamine and muramic acid; Benner and Kaiser, 2003), uronic acids (glucuronic
and galacturonic acids; Hung et al., 2003; Engel and Handel, 2011), methylated and dimethylated
sugars (Panagiotopoulos et al., 2013), heptoses (Panagiotopoulos et al., 2013) and sugar alcohols
(Van Pinxteren et al., 2012).

Free monosaccharide concentrations range from 10 to100 nM; they account < 10% of total

dissolved neutral sugars (TDNS), and experiments have shown that they are rapidly utilized
(minutes to hours) by bacterioplankton and as such they are considered as labile organic matter
(Rich et al., 1996; Skoog et al., 1999; Kirchman et al., 2001). Polysaccharide or dissolved combined
neutral sugars (DCNS) concentrations range from 200-800 nM; they account for 80-95% of TDNS
and experiments have shown that they disappear within time scales of days to months and, as such,
they are considered as labile and semi-labile organic matter (Aluwihare and Repeta, 1999; Carlson
and Hansell, 2015 and references therein). Other studies have shown that this labile and/or semi-
labile organic matter accumulates in the surface ocean and may potentially be exported to depth
contributing to the ocean carbon pump (Goldberg et al., 2010; Carlson and Hansell, 2015).

In the frame of the OUTPACE project we studied DOM dynamics in terms of DOC and DCNS

composition and tried to evaluate their residence time. The results are presented and discussed
along with heterotrophic prokaryotic production in order to better understand the bacterial cycling
of DOM  in the region.

**2. Materials and Methods**
**2.1 Sampling**
Sampling took place along a 5500 Km transect spanning from New Caledonia to French
Polynesia in the WTSP aboard the R/V *L'Atalante* during the Oligotrophy to Ultraoligotrophy
Pacific Experiment (OUTPACE) cruise (19 February-5 April, 2015). Samples were taken from 18
different stations comprising three long duration stations (LDA, LDB, and LDC; about 7-8 days)
and 15 short duration (SD1-15) stations (~8 h). Biogeochemical and physical characteristics of
these sites are described in detail elsewhere (Moutin et al., 2017). Briefly, the cruise took place
between 18-20°S covering two contrasted trophic regimes with increasing oligotrophy from west to
east (Fig. 1).
Discrete seawater samples were collected from 12 L Niskin bottles equipped with Viton O-rings
and silicon tubes to avoid chemical contamination. For DOC and DCNS analyses, samples were
filtered through two pre-combusted (450°C for 24 h) GF/F filters using a custom-made all-
glass/Teflon filtration syringe system. Samples for DOC (SD: 1-15 including LD: A, B ,C) were
collected into precombusted glass ampoules (450°C, 6h) that were sealed after acidification with
$H_3PO_4$ (85%) and stored in the dark at 4°C. Samples for DCNS (SD 1, 3-7, 9, 11, 13-15 including
LD: C) were collected in 40-mL Falcon vials (previously cleaned with 10% of HCl and Milli-Q
water) and frozen at -20°C until analysis.

**3. Chemical and microbiological analyses**
**3.1. Dissolved organic carbon (DOC) determination**
DOC was measured by high temperature combustion on a Shimadzu TOC-L analyzer (Cauwet,
1999). Typical analytical precision was ± 0.1-0.5 µM C (SD) for multiple injections (3-4) of
replicate samples. Consensus reference materials were injected every 12 to 17 samples to ensure
stable operating conditions and were in the range 42-45 µM (lot # 07-14;
(http://yyy.rsmas.miami.edu/groups/biogeochem/Table1.html).

**3.2. Dissolved combined neutral sugars (DCNS) determination**
*3.2.1. Carbohydrate extraction and isolation*
Seawater samples were desalted using dialysis tubes with a molecular weight cut-off of 100-500
Da (Spectra/Por® Biotech cellulose ester) according to the protocol of Panagiotopoulos et al.
(2014). Briefly, the dialysis tube was filled with 8 mL of the sample and the dialysis was conducted
into a 1 L beaker filled with Milli-Q water at 4°C in the dark. Dialysis was achieved after 4-5 h
(salinity dropped from 35 to 1-2 g L$^{-1}$). Samples were transferred into 40 mL plastic vials (Falcon;
previously cleaned with 10% HCl and Milli-Q water), frozen at -30 °C, and freeze dried. The
obtained powder was hydrolyzed with 1M HCl for 20 h at 100°C and the samples were again freeze
dried to remove the HCl acid (Murrell and Hollibaugh, 2000; Engel and Handel, 2011). The dried
samples were diluted in 4 mL of Milli-Q water, filtered through quartz wool, and pipetted into
scintillation vials for liquid chromatographic analysis. The vials were kept at 4°C until the time of
analysis (this never exceeded 24 h). The recovery yields of the whole procedure (dialysis and
hydrolysis) were estimated using standard polysaccharides (laminarin, and chondroitine sulfate) and
ranged from 82 to 86% (n=3). Finally, it is important to note that the current desalination procedure
does not allow the determination of the dissolved free neutral sugars (i.e., sugar monomers present
in samples with MW ~ 180 Da) because these compounds are lost/poorly recovered during the
dialysis step (Panagiotopoulos et al., 2014).

*3.2.2. Liquid Chromatography*
Carbohydrate concentrations in samples were measured by liquid chromatography according to
Mopper et al. (1992) modified by Panagiotopoulos et al. (2001, 2014). Briefly, neutral
monosaccharides were separated on an-anion exchange column (Carbopac PA-1, Thermo) by
isocratic elution (mobile phase 19 mM NaOH) and were detected by an electrochemical detector set
in the pulsed amperometric mode (Panagiotopoulos et al., 2014). The flow rate and the column
temperature were set at 0.7 mL min$^{-1}$ and 17°C, respectively. Data acquisition and processing were
performed using the Dionex software Chromeleon. Repeated injections (n = 6) of a dissolved
sample resulted in a CV of 12-15% for the peak area, for all carbohydrates. Adonitol was used as an
internal standard and was recovered at a percentage of 80-95%; however, we have chosen not to
correct our original data.

**3.3. Bacterial production**

Heterotrophic prokaryotic production (here abbreviated classically as "bacterial" production
or BP) was determined onboard with the $^{3}$H-leucine incorporation technique to measure protein
synthesis (Smith and Azam, 1992). Additional details are given in Van Wambeke et al. (2018).
Briefly, 1.5 mL samples were incubated in the dark for 1-2 h after addition of $^{3}$H leucine, at a final
concentration of 20 nM, with standard deviation of the triplicate measurements being on average
9%. Isotopic dilution was checked and was close to 1 (Van Wambeke et al, 2018), and we therefore
applied a conversion factor of 1.5 Kg C mol leucine$^{-1}$ to convert leucine incorporation to carbon
equivalents (Kirchman, 1993). BP was corrected for leucine assimilation by *Prochlorococcus*
(Duhamel et al., 2018) as described in Van Wambeke et al. (2018). To estimate bacterial carbon
demand (BCD) which is used to calculate semi-labile DOC residence time, we used a bacterial
growth efficiency (BGE) of 8% as determined experimentally using dilution experiments during the
OUTPACE cruise (Van Wambeke et al., 2018). BCD was calculated by dividing BP values at each
station by BGE.  Euphotic zone integrals were then computed from volumetric rates.

**4. Results**

4.1 General observations

The OUTPACE cruise was conducted under strong stratification conditions (Moutin et al.,
2018) during the austral summer encompassing a longitudinal gradient starting at the oligotrophic
Melanesian Archipelago (MA waters; stations SD1-SD12 including LDA and LDB stations) and
ending in the ultra-oligotrophic western part of the South Pacific gyre (WGY waters; stations SD13-
SD15 including LDC station; Fig. 1). Additional information on the hydrological conditions of the
study area (*i.e* temperature, salinity) including water masses characteristics is provided elsewhere
(de Verneil et al., 2018; Moutin et al., 2018). Mixed layer depth ranged from 11 to 34 m with higher
values recorded in the WGY (Moutin et al., 2018). The depth of the deep chlorophyll maximum
ranged from 69 to 119 m and from 122 to 155 m for the MA and WGY areas, respectively. Two
different trends can be noticed in a first approach:
a. Most of the biogeochemical parameters examined in the OUTPACE cruise (chlorophyll $\alpha$
concentrations, primary production, BP, BCD, $N_2$ fixation rates, and nutrient concentrations)
showed significantly higher values in the MA area than in the WGY area (Moutin et al., 2018; Van
Wambeke et al., 2018; Benavides et al., 2018; Caffin et al., 2018). These differences were also
reflected by the distribution of the diazotrophic communities detected in both areas further
highlighting the different dynamics across the oligotrophic gradient (Stenegren et al., 2018; Moutin
et al., 2017, 2018). The net heterotrophic/autotrophic status of the MA and WGY areas has been
discussed in previous investigations by comparing BCD and gross primary production (GPP) (Fig.
2). By using propagation of errors, Van Wambeke et al. (2018) concluded that GPP minus BCD
could not be considered different from zero at most of the stations investigated (11 out of 17)
showing a metabolic balance. For the other stations, net heterotrophy was shown at stations SD 4, 5,
6 and LDB, and net autotrophy at station SD9 (Van Wambeke et al, 2018).

b. The bulk of DOM as shown by DOC analysis did not follow the above biogeochemical

pattern and showed little variability on DOC absolute concentrations although a deeper penetration
of DOM was noticeable at 150 m depth in the WGY area (Fig. 3a; Table 1). As such, epipelagic (0-
200 m) DOC concentrations throughout the OUTPACE cruise ranged from 47 to 81 µM C (mean ±
sd: 67 ± 10 µM; n = 136) except at LDB (~85 µM C) which is probably related to a decaying
phytoplankton bloom (de Verneuil et al., 2018; Van Wambeke et al., 2018). Mesopelagic (200-1000
m) DOC values varied between 36 to 53 µM C (mean ± sd: 46 ± 4 µM; n = 67) (Fig. 4a; Table 1)
and are in agreement with previous studies in the South Pacific Ocean (Doval and Hansell, 2000;
Hansell et al., 2009; Raimbault et al. 2008).

DCNS concentrations closely followed DOC trends and fluctuated between 0.2-4.2 µM C

(mean ± sd: 1.9 ± 0.8 µM; n = 132) in the epipelagic zone (Fig. 3b; Table 1). These values are in
good agreement to those previously reported for the central and/or the eastern part of the South
Pacific gyre (1.1-3.0 µM C; Sempéré et al., 2008) that were recorded under strong stratification
conditions during austral summer (Claustre et al., 2008). Compared with other oceanic provinces
our epipelagic DCNS concentrations fall within the same range of those reported in the BATS
station in the Sargasso Sea (1.0-2.7 µM C) also monitored under stratification conditions (Goldberg
et al., 2010). Mesopelagic DCNS concentrations ranged from 0.3 to 2.4 µMC (average ± sd: 1.2 ±
0.6 µM; n = 68) (Fig. 4b; Table 1) and concur with previously reported literature values at the
ALOHA station (0.2-0.8 µMC; Kaiser and Benner, 2009) or in the Equatorial Pacific (0.8-1 µMC;
Skoog and Benner, 1997).

4.2 DCNS yields and composition

The contribution of DCNS-C to the DOC pool is referred to here as DCNS yields and is
presented as a percentage of DOC (*i.e* DCNS-C x $DOC^{-1}$ %). Epipelagic (0-200 m) average DCNS
yields, based on volumetric data, were similar between the WGY (range 0.3-5.1%; average ± sd: 2.8
± 1.3%; n = 41) and MA (range 0.8-7.0%; average ± sd: 2.8 ± 1.0%; n = 91) areas whereas deeper
than 200 m they were 2.4 ± 1.0% (n = 23) and 2.7 ± 1.3% (n = 43)  for the WGY and MA,
respectively (Table 1). These values are in good agreement to those reported for the eastern part of
the gyre (Sempéré et al., 2008) and concur well with the range of values (2-7%) recorded in the
Equatorial Pacific (Rich et al., 1996; Skoog and Benner, 1997).
The molecular composition of carbohydrates revealed that glucose was the major
monosaccharide at all depths in both the MA and WGY areas accounting on average for 53 ± 18%
(n = 132) of the DCNS in epipelagic waters and 64 ± 21% (n = 68) in mesopelagic waters (Table 1).
Epipelagic glucose concentrations (DCGlc-C) averaged 1.0 ± 0.6; n = 132 in both areas (Fig. 3c,
Table 1), however, a significantly higher mol% contribution of glucose was recorded in the WGY
than the MA especially at depths > 200 m (Fig. 5). Glucose was followed by xylose (9-12%),
galactose (4-9%) and mannose (5-8%) whereas the other monosaccharides accounted for < 6% of
DCNS (Fig. 5). The same suite of monosaccharides was also reported by Sempéré et al. (2008)
although the latter author also found that arabinose was among the major monosaccharides. Finally,
it is worth noting that the relative abundance of glucose increased with depth and sometimes
accounted 100% of the DCNS (Table 1, Fig. 5).

4.3 DOC and DCNS integrated stocks

DOC stocks (euphotic zone integrated) were calculated at the same stations where carbohydrate
(DCNS) data were available and were compared between the MA (stations: SD 1, 3, 4, 5, 6, 7, 9,
11) and WGY (SD13-SD15; LDC) stations (Fig. 6). DOC stock values in the euphotic were 9111 ±
1159 (n = 8) and 13266 ± 821 (n = 4) mmol C m$^{-2}$ for the MA and WGY areas, respectively. Excess
DOC stock (DOC$_{EX}$) was calculated by subtracting an average deep DOC value from the bulk
surface DOC pool. This DOC value was 40 µMC and was estimated averaging all DOC values
below 1000 m depth from all stations (39.6 ± 1.4 µMC, n = 36). DOC$_{EX}$ stock values averaged
3717 ± 528 (n = 8) and 5265± 301 (n = 4) mmol C m$^{-2}$ accounting about 40% of DOC in both areas.
DCNS represented 6.7 and 7.1% of DOC$_{ex}$ in the MA and WGY sites, respectively, further
suggesting that only a small percentage of DOC$_{EX}$ can be attributed to DCNS (polysaccharides).

**5. Discussion**

5.1 DOC and DCNS stocks in relation with biological activity

Euphotic zone integrated stocks of DOC, DOC$_{EX}$ and DCNS were respectively 46, 42 and 52%
higher in the WGY than in the MA (Fig. 6), as opposed to BCD and GPP (Fig. 2). This is a
consequence of the deepening of the euphotic zone, because the variability of the volumetric stocks
was high, and not statistically different in the euphotic zone between MA and WGY areas. As
indicated above DOC$_{EX}$ is calculated as the difference between the bulk surface DOC and deep
DOC the latter assumed to be refractory. Thus, DOC$_{EX}$ is often described as "semi-labile" DOC or
DOC$_{SL}$ with a turnover on time scales of weeks to months (Carlson and Hansell, 2015). DCNS
belong to this semi-labile category of DOC (Biersmith and Benner, 1998; Aluwihare et Repeta,
1999; Benner, 2002), and the results of this study showed that DCNS represented a low proportion
(~7%) of DOC$_{EX}$. Because the conditions of the HPLC technique employed in this study does not
allow identification and quantification of all the carbohydrate components of DOC (methylated
sugars, uronic acids, amino sugars etc) it is possible that the contribution of polysaccharides to the
DOC$_{EX}$ is underestimated. Previous investigations on amino sugars and methylated sugars indicated
that these monosaccharides account for < 3% of the carbohydrate pool (Benner and Kaiser;
Panagiotopoulos et al., 2013) while uronic acids may account for as much as 40% of the
carbohydrate pool (Engel et al., 2012) indicating that the latter compounds should at least be
considered in future DOM lability studies.

Other semi-labile compounds that potentially may contribute to the $DOC_{EX}$ pool are proteins

and lipids. Unfortunately, proteins (combined amino acids) were not measured in this study.
Nonetheless, previous investigations indicated that total dissolved amino acids represent 0.7-1.1%
of DOC in the upper mesopelagic zone of the north Pacific (Kaiser and Benner, 2012) further
suggesting a relatively small contribution of amino acids to the $DOC_{EX}$. During the OUTPACE
cruise, assimilation rates of $^3H$- leucine using concentration kinetics were determined (Duhamel et
al., 2018) and, based on the Wright and Hobbie (1966) protocol, the ambient concentration of
leucine was determined. The results showed a lower ambient leucine concentration at the LDC
(0.56 nM) than at the LDA (1.80 nM) stations (Duhamel et al., 2018).

This result may suggest that single amino acid and perhaps proteins concentrations are very low

at the LDC station, reflecting the ultra- oligotrophic regime of the WGY. On the other hand, DOM
exhibited only slightly different C/N ratios between MA (C/N = 13) and WGY (C/N =14), which
does not suggest differences in DON dynamics in relation with organic matter lability (data from
integrated values of 0-70 m; Moutin et al., 2018). Clearly further investigations are warranted on
combined and free amino acids distribution in relation with $N_2$ fixation.

The high stock of $DOC_{EX}$ measured in WGY was also characterized by an elevated residence

time ($T_{r SL}$) calculated as the ratio of $DOC_{EX}$ / BCD. This ratio is calculated based on the
assumption that $DOC_{EX}$ is representative of the $DOC_{SL}$ and the latter  pool turnover is at the scale
of seasonal mixing (i.e weeks to months) whereas the BP,  as determined with leucine technique on
short incubation times (1-2 hours), tracks only the ultra-labile to labile organic matter consumption
and not $DOC_{SL}$ utilization. Biodegradation experiments (3 experiments, duration 10 days each)
performed during the OUTPACE cruise showed that the labile DOC represented only 2.5 to 5% of

the DOC pool (Van Wambeke et al., 2018), confirming that the residence time calculated from $DOC_{EX}$ / BCD overestimates the residence time of ultra-labile DOC. The bacterial production and BGEs associated with the use of semi-labile DOC is currently not technically measurable due to long-term confinement artifacts. Our results showed that $T_{r\,SL}$ in the WGY was in the order of 176 ± 43 days (n = 3), i.e. about three times higher than in the MA region ($T_{r\,SL}$= 51 ± 13 days (n = 8)) indicating an accumulation of the semi-labile DOM in the surface waters of WGY (Fig. 7). As suggested by previous studies the accumulation of DOC in the surface waters of oligotrophic regimes may be related in biotic and/or abiotic factors.

Nutrient limitation can prevent DOC assimilation by heterotrophic bacteria and as such sources and sinks are uncoupled, allow accumulation (Thingstad et al., 1997; Jiao et al., 2010; Shen et al., 2016). Biodegradation experiments (Van Wambeke et al., 2018) focusing on the determination of the BGE and the degradation of the labile DOC pool (turning over 10 days) revealed a less biodegradable DOM fraction and lower degradation rates at the LDC (2.4% labile DOC; 0.012 $d^{-1}$) than the LDA site (5.3% labile DOC; 0.039 $d^{-1}$). Other experiments, focusing on the factors limiting BP by testing the effect of different nutrient additions, showed that over a short-time period, BP is initially limited by the availability of labile carbon in the WGY (as tracked with glucose addition, Van Wambeke et al., 2018). This limitation on BP by labile carbon/energy was also the case at the center of the South Pacific gyre (Van Wambeke et al., 2008), while N limitation (as tracked by addition of ammonium+nitrate) was more pronounced in the MA area.

Although extensive photodegradation may transform recalcitrant organic matter into labile, the low content in chromophoric DOM recorded in the surface waters of WGY ($a$CDOM(350) = 0.010-0.015 $m^{-1}$, 0-50 m; Dupouy et al. unpublished results from the OUTPACE cruise) points toward an already photobleached and thus photodegraded organic material (Tedetti et al., 2007; Carlson and Hansel, 2015). Notably, the 10% irradiance depths for solar radiations (Z 10%) clearly showed a higher penetration of UV-R and PAR radiations in the WGY area than in MA area (Dupouy et al.,

2018). These results are in agreement with previous investigations reporting intense solar radiation in the South Pacific gyre highlighting an strong decrease of chromophoric dissolved organic matter (CDOM) in the gyre (Tedetti et al., 2007). Less energy available for heterotrophic prokaryotes should prevent them from degrading such recalcitrant, photo-degraded organic matter.

The computation of the carbon, nitrogen, and phosphorus budgets in the upper 0-70 m layer by Moutin et al. (2018) suggested that at 70 m the environmental conditions remained seasonally unchanged during the OUTPACE cruise, forming an average wintertime depth of the mixed layer. These authors calculated seasonal (from winter to austral summer) net DOM and POM accumulation on the basis of such assumptions, and found a dominance of DOC accumulation in the MA area (391 to 445 mmol m$^{-2}$ over 8 months). This DOC accumulation in the MA area was 3.8 to 8.1 times higher than that of POC accumulation during the same time period. On the other hand, only DOC accumulated at WGY, although the amount was two times lower in magnitude than in the MA (391- 445 vs 220 mmol m$^{-2}$). The accumulation of DOC and $DOC_{EX}$ (Fig. 6) in the WGY may have important implications with regard to the sequestration of this organic material in the mesopelagic layers. DOC appears to be the major form of export of carbon in the WGY area and this result agrees with the general feature observed in oligotrophic regimes (Roshan and Devries, 2017).

5.2 DCNS dynamics across the South West Pacific

Previous investigations have employed the DCNS yields along with mol% of glucose to assess the diagenetically "freshness" of organic matter (Skoog and Benner, 1997; Benner, 2002; Goldberg et al. 2010). In general freshly produced DOM has DCNS yields >10% and mol% glucose between 28-71% (Biersmith and Benner, 1998; Hama and Yanagi, 2001). Elevated mol% glucose (> 25%) does not necessarily mirror fresh material because such values have also been reported for deep

DOM and low molecular weight DOM that are considered as a diagenetically altered material
(Skoog et al., 1997).

Our results showed that epipelagic DCNS yields were about similar (~2.8%) in both WGY and

MA areas (Table 1) further indicating a similar contribution of DCNS to the DOC pool despite the
major differences observed for the other biochemical parameters (e.g. deepening of the nitraclines
and deep chlorophyll maximum etc) between MA and WGY. As expected, DCNS yields decreased
by depth but were always comparable between WGY and MA areas (Table 1). By analogy to the
$DOC_{SL}$, we tried to estimate a DNCS residence time assuming that (a) the ectoenzymatic hydrolysis
is a rate-limiting step for bacterial production, ii) the mean contribution of polysaccharides
hydrolysis to bacterial production is 11%, based on Pointek et al. (2011), and iii) this 11%
correction factor can be propagated to BCD. On the basis of these assumptions, we estimated a
DCNS residence time as DCNS/(11% x BCD). The results showed that DCNS exhibited a higher
residence time in the WGY ($T_{r\,DCNS-C}$= 91 ± 41 days, n = 3) than the MA area ($T_{r\,DCNS-C}$ = 31 ± 10
days, n = 8) which clearly shows that the DCNS pool persist longer in the surface waters of the
WGY (Fig. 7). Moreover, because carbohydrates do not absorb light these polysaccharides (DCNS)
do not seem to be impacted by the high photochemistry in WGY and potentially may be exported in
the Ocean interior during a non-stratification period (e.g. winter time) considering their high
residence time at the WGY area. In addition, their slow utilization could also be related to energy
limitation by heterotrophic prokaryotes in the WGY area.

Glucose accounted for ~50% of DCNS in the MA surface waters which most likely reflects the

high abundance of *Trichodesmium* species in that area (Dupouy et al., 2018; Rousset et al., 2018). A
roughly similar percentage of glucose was also recorded in surface WGY waters (Fig. 5a) which is
probably due to the low utilization of semi-labile organic matter in the form of exopolysaccharides.
These exopolysaccharides are probably hydrolyzed by bacteria, but not taken up due to limited
nutrient availability. At 200 m depth, glucose accounted for 75% and 50% of DCNS in the WGY

and MA areas, respectively (200 m depth), and this percentage increased considerably with depth in both areas (76% for MA and 96% for WGY at 2000 m depth) indicating a preferential removal of the other carbohydrates relative to glucose (Fig. 5b; Fig. 5c). The low DCNS yields (~1%) at 2000 m depth along with the high % mol abundance of glucose clearly suggests the presence of diagenetically altered DOM and is consistent with previous investigations (Skoog and Benner, 1997; Goldberg et al. 2010; Golberg et al., 2011).

**6. Conclusions**

This study showed a rather uniform distribution of DOC and DCNS concentrations in surface waters across an increasing oligotrophic gradient in the South West Pacific Ocean during the OUTPACE cruise. Nevertheless, our results showed that DOC and $DOC_{EX}$ stocks were by ~40% in WGY than the MA area, accompanied with higher residence times in the WGY area suggesting an accumulation of semi-labile material in the euphotic zone of WGY. Although DCNS accounted a small fraction of $DOC_{SL}$ (~7%) our results showed that DCNS or polysaccharides also exhibited a higher residence time ($T_{r\ DCNS-C}$) in the WGY than in the MA area indicating that DCNS persist longer in the WGY. This $T_{r\ DCNS-C}$ is calculated on the basis of many assumptions on DNCS hydrolysis rates that were not practically determined, showing the need to estimate such fluxes in order to better estimate the dynamics of carbohydrates. Glucose was the major monosaccharide in both areas (51 - 55%) and its relative abundance increased with depth along with a decrease of the DCNS yields indicating a preferential removal of the other carbohydrates relative to glucose. Clearly further investigations are warranted to better characterize the semi-labile DOC pool in terms of combined and free amino acids distribution in relation with $N_2$ fixation.

**Acknowledgements**

This is a contribution of the OUTPACE (Oligotrophy from Ultra-oligoTrophy PACific Experiment) project lead by T. Moutin and S. Bonnet and funded by the French national research agency (ANR-14-CE01-0007-01), the LEFE-CyBER program (CNRS-INSU), the GOPS program (IRD) and CNES (BC T23, ZBC 4500048836). The OUTPACE cruise (http://dx.doi.org/10.17600/15000900) was managed by the MIO from Marseille (France). The authors thank the crew of the R/V L'Atalante for outstanding shipboard operation. G. Rougier and M. Picheral are thanked for their efficient help in CTD rosette management and data processing. C. Schmechtig is acknowledged for the LEFE CYBER database management. We also thank A. Lozingot for administrative aid for the OUTPACE project. The authors also acknowledge Prof. R. Benner and one anonymous reviewer for valuable comments and fruitful discussions. M.B. was funded by the People Programme (Marie Skłodowska-Curie Actions) of the European Union's Seventh Framework Programme (FP7/2007-2013) under REA grant agreement number 625185. C.P. received support from the PACA region (MANDARINE project, grant number 2008-10372) and Aix Marseille University (ORANGE project, FI-2011).

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

**Figure and Table captions:**

Figure 1: Sampling stations during the OUTPACE cruise. The white line shows the vessel course (data from the hull-mounted ADCP positioning system). Stations and their respective names (SD1-SD15 including LDA, LDB and LDC) are depicted in grey. Figure courtesy of T. Wagener.

Figure 2: Integrated stocks of bacterial carbon demand (BCD) and gross primary production (GPP) (mmol C m$^{-2}$ d$^{-1}$) over the euphotic zone. Data from Van Wambeke et al. (2018). Error bars

Figure 3: Distribution of A: dissolved organic carbon (DOC); B: dissolved combined neutral sugars

(DCNS); and C: dissolved combined glucose (DCGlc) in the upper surface layer (0-200 m) of the

study area. DCNS and DCGlc concentration is given in carbon equivalents in order to have the

same unit as DOC. Long duration stations (LDA, LDB and LDC) are also indicated in each graph.

White and red circles indicate the mixed layer depth and deep chlorophyll maximum, respectively

for each station.

Figure 4: Depth profiles of A: DOC; B: DCNS; and C: DCGlc in the 0-2000 m layer of the study

area.

Figure 5: Average Mol percentage (mol %) of dissolved monosaccharides at A: surface; B: 200 m;

and C: 2000 m depth for MA and WGY areas. Monosaccharides abbreviations: Fuc.: Fucose;

Rha.:Rhamnose; Ara.: Arabinose; GlcN.: Glucosamine; Gal.: Galactose; Glc.: Glucose; Man.:

Mannose and Xyl.: Xylose.

Figure 6: Integrated carbon stocks (mmol C m$^{-2}$) over the euphotic zone carbon in terms of DOC,

DOC$_{EX}$ and DCNS-C. * DOC and DOC$_{SL}$ were statistically different between MA and WGY areas

(Man-Whitney test, p<0.05).

Figure 7: Residence time (days) of semi labile DOC (T$_{r\ SL}$) and DCNS-C (T$_{r\ DCNS-C}$) for MA and

WGY areas. * T$_{r\ SL}$ and T$_{r\ DCNS-C}$ were statistically different between MA and WGY areas (Man-

Whitney test, p<0.05).

Table 1: Range and mean values (0-200 m and 200-1000 m) of DOC (µMC), DCNS-C (µMC),

DCGlc-C (µMC), DCNS-C/DOC (%) and DCGlc-C/DCNS-C (%) recorded during the OUTPACE

cruise. MA comprises the SD2-SD12 stations and WGY comprises the LDC and SD13-SD15.

Means of MA and WGY were not statistically different for any of the parameters presented (Man-

Whitney test, p > 0.05).

655

656

Table 1: Range and mean values (0-200 m and 200-1000 m) of DOC, DCNS-C, DCGlc-C, DCNS-C/DOC and DCGlc-C/DCNS-C recorded during the OUTPACE cruise. MA comprises the SD2-SD12 stations and WGY comprises the LDC and SD13-SD15. Means of MA and WGY were not statistically different for any of the parameters presented (Man-Whitney test, p >0.05).

| | All data | | | | MA | | | | WGY | | | |
|---|---|---|---|---|---|---|---|---|---|---|---|---|
| | Range | mean±sd (n) | Range | mean±sd (n) | Range | mean±sd (n) | Range | mean±sd (n) | Range | mean±sd (n) | Range | mean±sd (n) |
| DOC (µM) | 47-81 | 67±10 (136) | 36-53 | 46±4 (67) | 51-79 | 66±9 (94) | 39-52 | 46±3 (43) | 47-81 | 68±10 (42) | 36-53 | 46±4 (24) |
| Depth (m) | 0-200 | | 200-1000 | | 0-200 | | 200-1000 | | 0-200 | | 200-1000 | |
| | | | | | | | | | | | | |
| DCNS-C (µM) | 0.2-4.2 | 1.9±0.8 (132) | 0.3-2.4 | 1.2 ±0.6 (68) | 0.6-4.2 | 1.8±0.7 (91) | 0.3-2.4 | 1.2±0.6 (45) | 0.2-3.8 | 1.9±1.0 (41) | 0.3-2.0 | 1.0±0.4 (23) |
| Depth (m) | 0-200 | | 200-1000 | | 0-200 | | 200-1000 | | 0-200 | | 200-1000 | |
| | | | | | | | | | | | | |
| DCGlc-C (µM) | 0.2-3.0 | 1.0±0.6 (132) | 0.2-1.6 | 0.7±0.3 (68) | 0.3-3.0 | 1.0±0.6 (91) | 0.2-1.6 | 0.7±0.4 (45) | 0.2-2.7 | 1.1±0.7 (41) | 0.3-1.4 | 0.7±0.3 (23) |
| Depth (m) | 0-200 | | 200-1000 | | 0-200 | | 200-1000 | | 0-200 | | 200-1000 | |
| | | | | | | | | | | | | |
| DCNS-C/DOC (%) | 0.3-7.0 | 2.8±1.1 (132) | 0.56-5.4 | 2.6±1.2 (66) | 0.8-7.0 | 2.8±1.0 (91) | 0.6-5.4 | 2.7±1.3 (43) | 0.3-5.1 | 2.8±1.3 (41) | 0.6-4.7 | 2.4±1.0 (23) |
| Depth (m) | 0-200 | | 200-1000 | | 0-200 | | 200-1000 | | 0-200 | | 200-1000 | |
| | | | | | | | | | | | | |
| DCGlc-C/DCNS-C (%) | 19-100 | 53±18 (132) | 35-100 | 64±21 (68) | 28-100 | 54±17 (91) | 36-100 | 63±22 (45) | 19-100 | 58±20 (41) | 35-100 | 66±20 (23) |
| Depth (m) | 0-200 | | 200-1000 | | 0-200 | | 200-1000 | | 0-200 | | 200-1000 | |

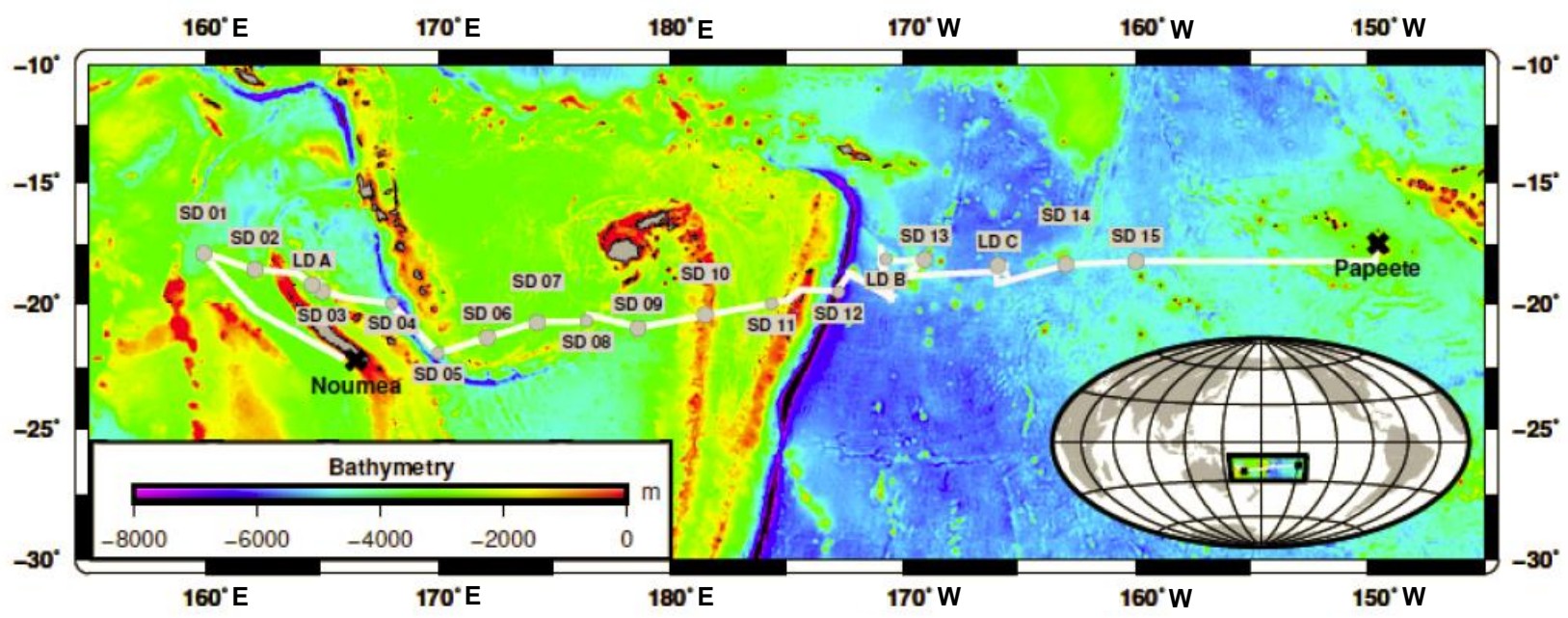

**Figure 1**

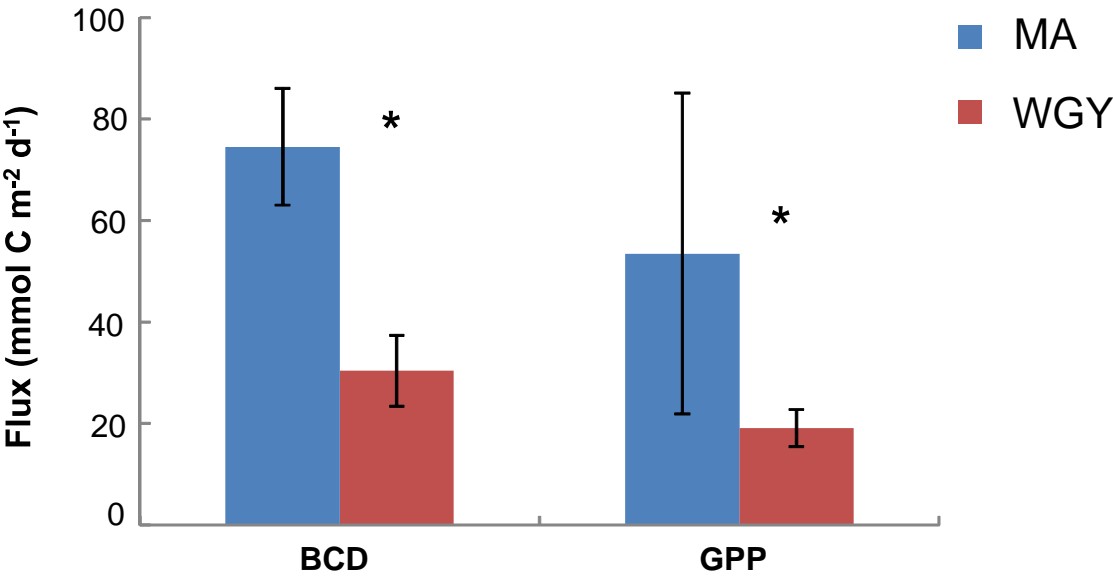

**Figure 2**

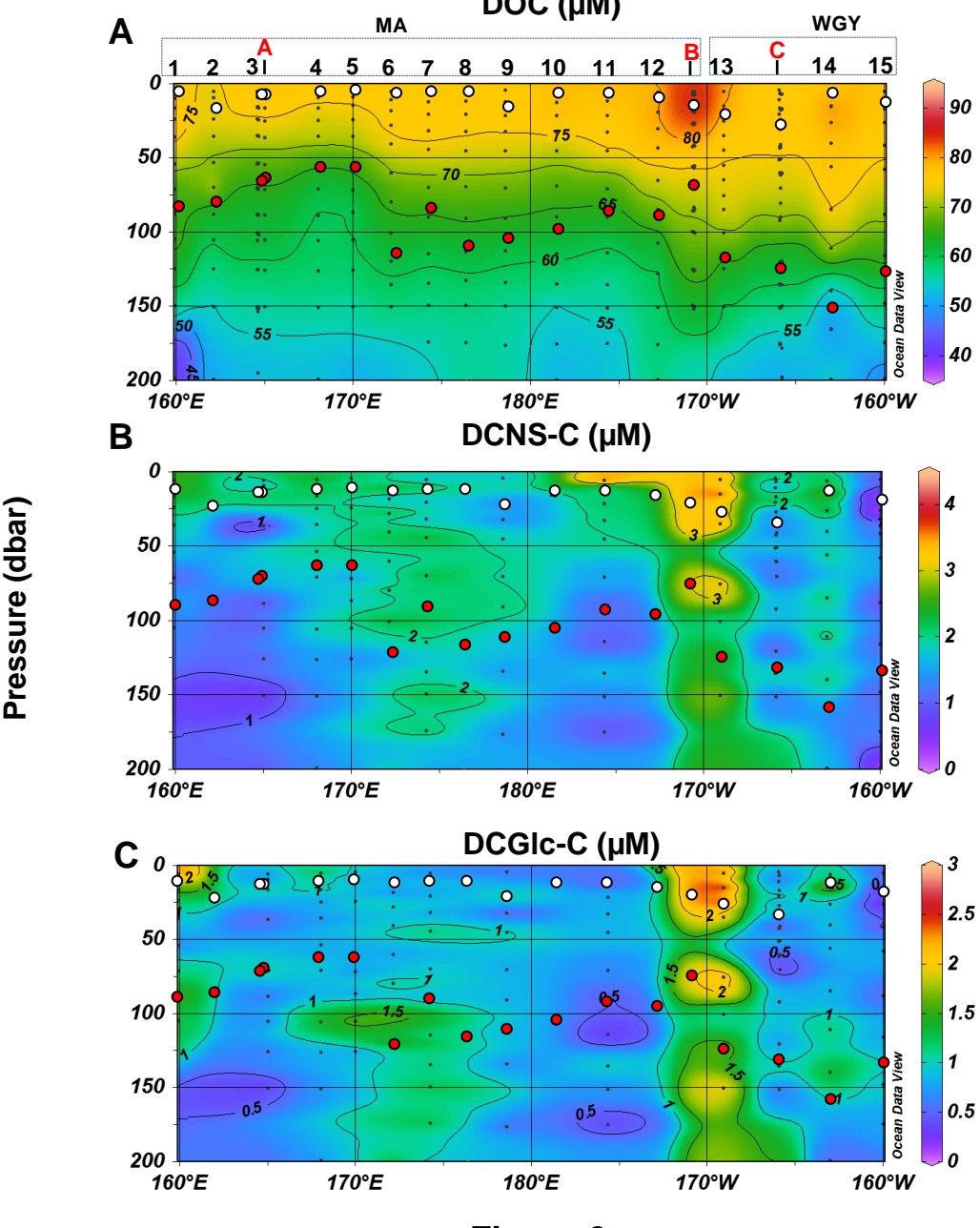

**Figure 3**

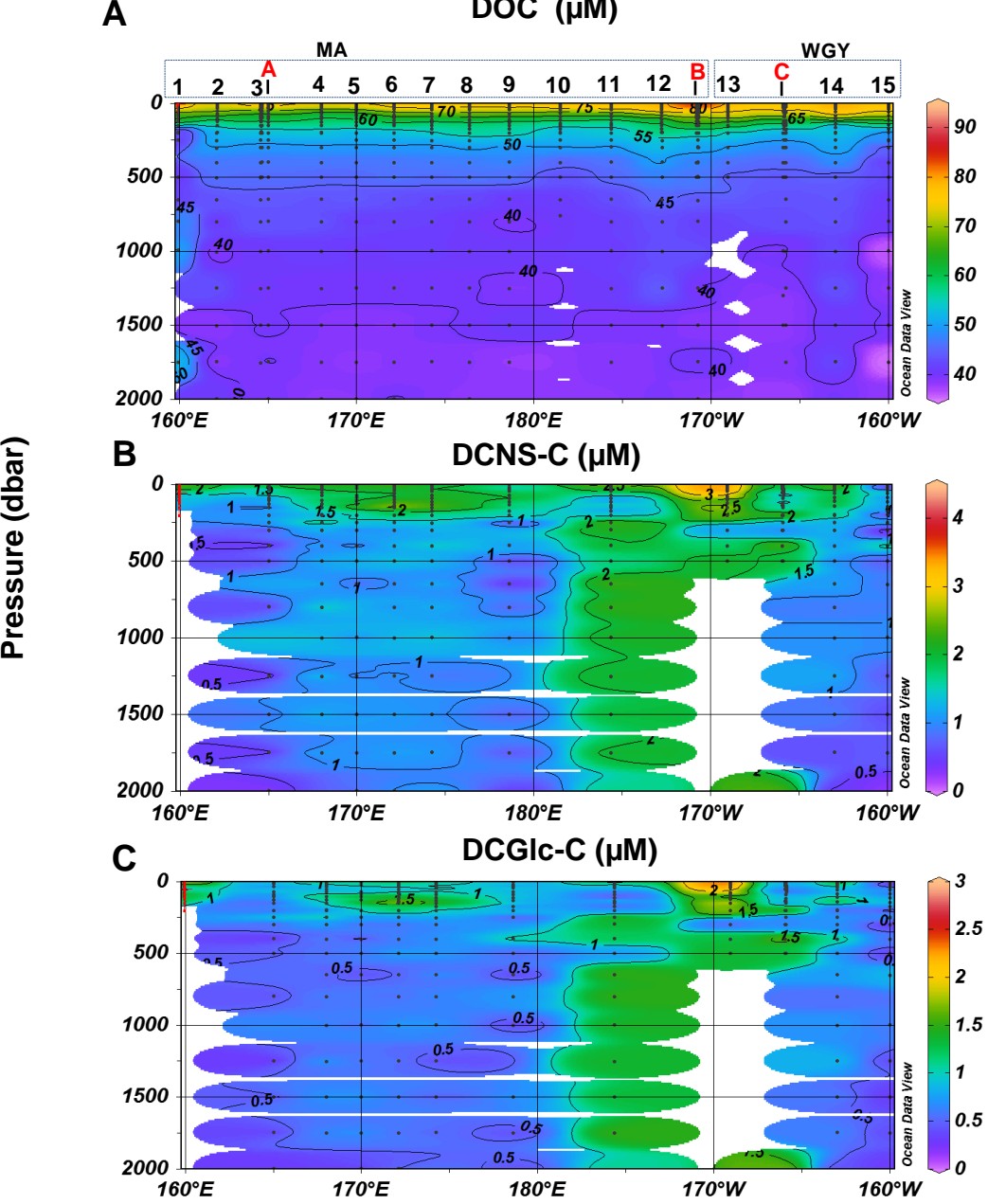

**Figure 4**

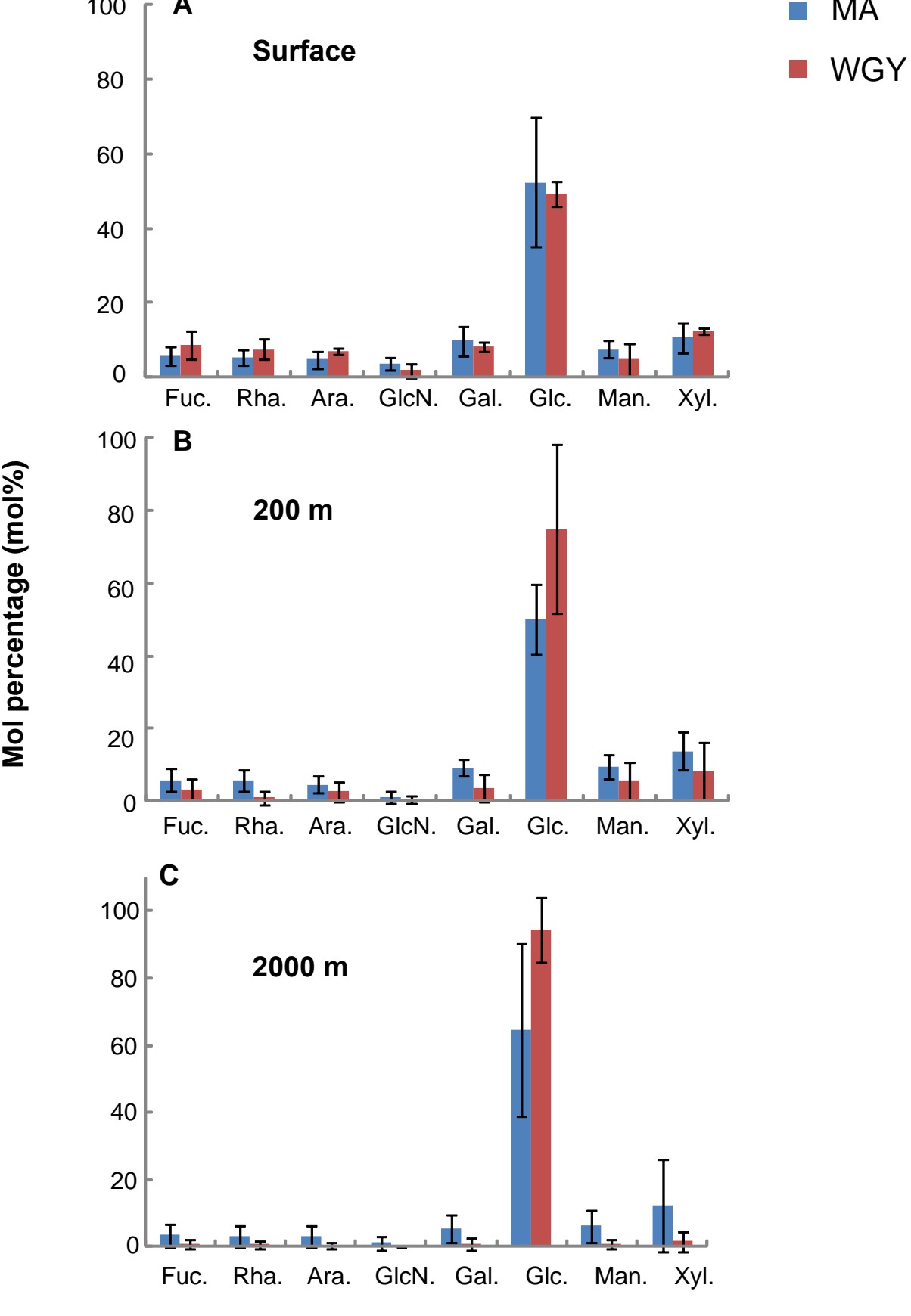

**Figure 5**

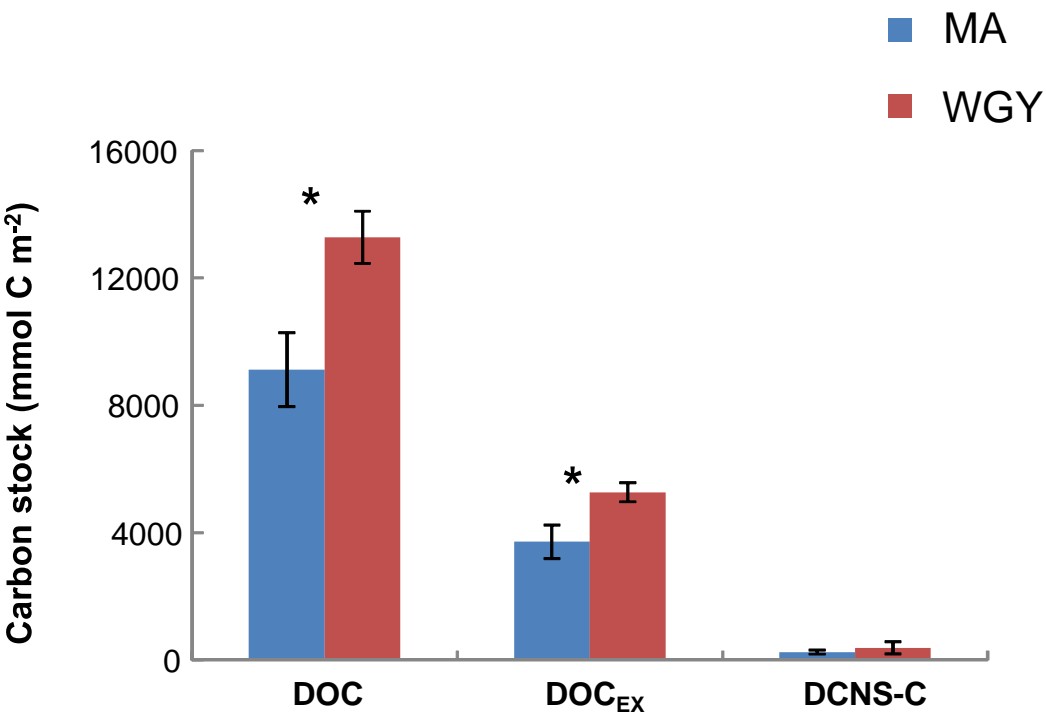

**Figure 6**

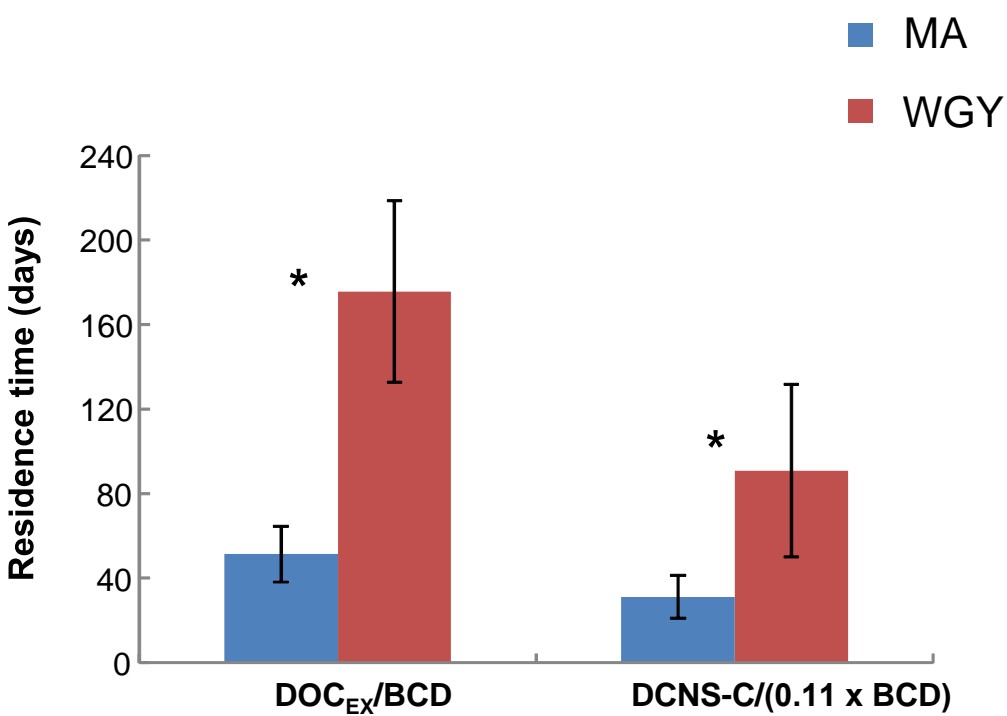

**Figure 7**