# Peer review of "The composition and distribution of semi-labile dissolved organic matter across the South West Pacific"

_Biogeosciences, 2018_

## Referee Comment (RC1) · R. Benner (Referee) · 9 May 2018

This study presents novel DOC and dissolved carbohydrate data along a transect in the South Pacific that spans a range of oligotrophic waters as part of the OUTPACE project. This is an understudied region of the global ocean, and these data provide interesting insights about carbohydrate dynamics in these oligotrophic waters. The study also provides a valuable portrait of DOC and carbohydrates for comparison to other ocean basins.

The carbohydrate reservoir in the ocean is large and molecularly diverse. Carbohydrates account for about 15-20% of marine DOC and are among the most abundant

biochemicals in seawater (Benner et al. 1992; Pakulski and Benner 1994; Goldberg et al. 2010). Neutral sugars were measured in this study and while they are important and relatively abundant carbohydrates, they are not the only carbohydrates in seawater. This is implied in the Introduction (lines 68-71) and in the first paragraph of the Discussion. The manuscript needs revision to clarify the diversity and abundance of carbohydrates in the ocean and to place neutral sugars within the broader carbohydrate reservoir. For example, many carbohydrates besides neutral sugars contribute to semi-labile DOC (lines 238-241).

The reported DCNS %DOC values in this study are similar to those observed in the N. Atlantic and N. Pacific (Goldberg et al. 2010, 2011; Kaiser and Benner 2009). This suggests carbohydrates are of a similar diagenetic state among these major ocean gyres. In contrast, the mol% glucose ($\sim$50-75%) values in this S. Pacific study are high compared to values reported ($\sim$20-50%) in the N. Atlantic and N. Pacific (Goldberg et al. 2010, 2011; Kaiser and Benner 2009). They are particularly high in surface waters ($\sim$50%). Is this indicative of a different source of carbohydrates in surface waters? Given the similar yields (%DOC) among ocean basins it seems unlikely the high mol% glucose in the S. Pacific is due to greater diagenetic processing. The authors need to address the high mol% glucose values in the Discussion section.

I recommend the authors include a table with depth, chlorophyll, [DOC], [DCNS], DCNS %DOC, and mol% glucose data from all stations in the upper 200 m of the water column. It is not possible to derive quantitative values from the figures (2, 3). The authors should take a look at the article by Shen et al. 2016 in L&O. This study shows the accumulation of carbohydrate-rich DOC during the summer in productive waters of the Gulf of Mexico. Nutrient limitation appears to play a role in carbohydrate-rich DOC accumulation.

Specific comments: Methods: It is unclear how Dissolved Combined Neutral Sugar (DCNS) is calculated. The DCNS terminology implies that free sugars have been subtracted from total sugars. If so, this should be stated. If not, the DCNS terminology

needs clarification because the term implies free sugars are not included. Clarification is important for comparison of values among studies using different terminologies. Line 31: change "also reflected" to "observed Line 34: delete "high" and change "higher" to "longer" Line 39: change "higher" to "longer" Line 49: change to "life" to "productivity " Line 72-73: The carbohydrate pool also includes oligosaccharides. Line 75: add reference McCarthy et al. 1996 Line 78: change "that" to "they" Line 79: delete "ultra" Line 116-118: Are these values for multiple injections of a single sample or replicate samples? Report the average ± SD concentration for the reference standard. Line 154: provide a reference for the carbon conversion factor Line 165-166: include the range of mixed layer depths observed along the transect Line 173: change "prokaryotic" to "bacterial" production (BP) Line 182-183: reported range (55-78) is inconsistent with the highest value being 85, add the median value along with the range Line 242-246: This sentence needs revision for clarity, e.g. 3H leucine concentrations. Duhamel et al. 2018 is not in the references. Total dissolved amino acids are known to contribute to semi-labile DOC and are measurable in mesopelagic waters of the Pacific. Line 194 and 196: add reference Kaiser and Benner 2009, values for the Pacific HOT station should be added for comparison Line 211: Fig. 5 is presented before Fig. 4. Figure 4 is first presented in the Discussion and should therefore be presented as Fig. 5. Line 251-252: Residence time (d) should be changed to Turnover time (d-1) because the calculation is based on microbial utilization (BCD) of DOC.

Fig. 1: The station symbols and numbers should be changed to colors that standout from the background. Fig. 2: The legend states data are presented from 0-300 m depth, but the figure shows 0-200 m depth. Fig. 3: The masking of data on figures 2B and 2C due to abnormal extrapolation of data is confusing and inappropriate. Rather than mask areas on the figures, these odv plots should be adjusted to properly extrapolate among profiles (i.e. not connected between profiles that are many kilometers apart). Fig. 4: Residence time (d) should be changed to Turnover time (d-1). Fig. 5 – The relative abundance of dissolved monosaccharides should be referred to as the mole percentage (mol %).

Additional editing for grammar would improve the text.

---

## Referee Comment (RC2) · Anonymous Referee #2 · 18 Jul 2018

The authors measured [DOC], [DCNS], and BP across a gradient of oligotrophic conditions in the South Pacific subtropical gyre. They calculate a variety of parameters relating to the role of bacteria in DOC cycling and carbon export in the region, including BCD and DOC residence times.

The results and discussion of this manuscript need to better reflect the data presented in the figures – there are some notable disconnects, such as BCD, where the results in the figures are neither presented in the Results section nor is the significance of those results expounded on in the discussion. I've noted the instances that most stand out to me here:

[Figure]

1) The Results section needs to include BP, BCD, and PP (the latter is perhaps from another study, but it could still be summarized). My first thought on looking at Figure 4, in context of the authors' discussion of accumulating DOCsl, was that it seemed like BCD and PP might be reversed. While I quickly realized that's not the case, as the manuscript stands, there is no way to double-check the figure against numerical values.

2) The discussion section needs to be expanded, as the context and significance of the results are largely glossed over. In particular, the authors neglect to discuss the incongruity of BCDs that so largely exceed PP, yet they note a residence time of DOCsl of up to several months and a glucose-heavy DCNS pool that implies highly worked-over DOM. So then what could possibly be supporting that high BCD in the gyre? Of course all of these measurements/estimations have uncertainties and assumptions, but those caveats need to be presented and/or the authors need to explain what they hypothesize is driving such extreme heterotrophy in the system.

I recommend the authors conduct one more proofread for grammar – by and large the manuscript is well written, but there are many instances of missing words such as prepositions.

Title: Would be more accurate to refer to DOC/DCNS rather than to labile DOM, as in fact the authors largely discuss presumed semi-labile DOM and make no direct measurements of DOM lability.

Abstract: I would find it useful to have a brief summary of methods in the abstract (i.e., that residence times were estimated by comparing stocks rather than by experimental approaches).

Line 59: The abbreviation of gyre alone is very unnecessary. It saves 2 letters but makes the text significantly less reader-friendly.

Line 67: I don't believe Goldberg et al. measured the percent of labile DOC consti-

tuted by carbohydrates, and further, the composition of most truly labile DOC isn't well characterized (e.g., reviewed by Carlson and Hansell 2015, cited in the manuscript).

Line 89: This sounds like a hypothesis, but there is no discussion of N2 fixation in the results. Even if characterizing the N2 fixation gradient was the overall cruise goal, it should be removed from this manuscript unless the authors were to actually compare their results with N2 fixation via correlations or similar.

Results: A table with results in numerical format needs to be included. I was interested in seeing real numbers to judge for myself if the 4% difference in DOC concentrations between regions was consistent enough to be significant. It's much more difficult for others to use the work as a reference in the future if they are limited to estimating values from averages in figures. Finally, this is bad practice for data availability purposes. Would be fine to put it in supplementary material as long as this is clearly referenced in the text.

Line 252: I would like to see more discussion of the caveats of this estimation of residence time. E.g., the semi-labile pool is heterogeneous and composed of compounds that will be more or less biologically available, while BCD is derived from a measurement of BP that lasted 1-2 hours and therefore is likely based on the use of the most labile compounds available at that time. This may still be a useful metric for comparing between the two regions but it needs to be presented less as a clear-cut value.

Lines 282-284: This sentence is confusing, please rephrase.

Line 303: How is this calculated? Is it DCNS divided by BCD, as for DOCsl above? That seems to make a lot of assumptions if so.

Line 306: The hypothesized role of carbohydrates in export to depth is not coherent with the statement three lines above that these compounds have residence times of 3 or 8 days, especially in a stratified system such as the gyre. Please expand on this statement to explain this speculation better. (It's obviously quite possible this is

happening, as DCNS are present at depth – perhaps this indicates an issue with the DCNS residence time calculation, as above?)

Line 324: I don't follow this sentence; please rephrase.

Line 541: Specify who/what is CLS.

Line 557: Carbon stock should not be d-1 here.

Figures 1-3: The longitude should be marked consistently between Fig. 1 and Figs. 2-3.

Figure 1: A locator map would be appreciated for those reading this as a stand-alone paper and not as part of the broader cruise special issue.

For Figs. 2 and 3, have you tried plotting panels B and C on the same axis? The ability to visually assess DCGlc as a proportion of DCNS might be worth the loss in resolution in panel C. (This is only a suggestion, please take it as such.)

Figure 5: It would help the reader to label the depths on each panel.

---

## Author Comment (AC1) · 1 Oct 2018

Reviewer #1: This study presents novel DOC and dissolved carbohydrate data along a transect in the South Pacific that spans a range of oligotrophic waters as part of the OUTPACE project. This is an understudied region of the global ocean, and these data provide interesting insights about carbohydrate dynamics in these oligotrophic waters. The study also provides a valuable portrait of DOC and carbohydrates for comparison to other ocean basins. The carbohydrate reservoir in the ocean is large and molecularly diverse. Carbohydrates account for about 15-20% of marine DOC and are among the most abundant biochemicals in seawater (Benner et al. 1992; Pakulski

and Benner 1994; Goldberg et al. 2010). Neutral sugars were measured in this study and while they are important and relatively abundant carbohydrates, they are not the only carbohydrates in seawater. This is implied in the Introduction (lines 68-71) and in the first paragraph of the Discussion. The manuscript needs revision to clarify the diversity and abundance of carbohydrates in the ocean and to place neutral sugars within the broader carbohydrate reservoir. For example, many carbohydrates besides neutral sugars contribute to semi-labile DOC (lines 238-241).

We agree with this comment and in the introduction of the revised MS we included a broader spectra of sugars including free monosaccharides (amino sugars, uronic acids, methylated sugars, sugar alcohols) reported in DOM or HMWDOM (see page 4, lines 79-83). We also expanded our discussion about the possible contribution of these compounds to the semi-labile DOC pool (see page 12, line 267-274).

The reported DCNS %DOC values in this study are similar to those observed in the N. Atlantic and N. Pacific (Goldberg et al. 2010, 2011; Kaiser and Benner 2009). This suggests carbohydrates are of a similar diagenetic state among these major ocean gyres. In contrast, the mol% glucose (_50-75%) values in this S. Pacific study are high compared to values reported (_20-50%) in the N. Atlantic and N. Pacific (Goldberg et al. 2010, 2011; Kaiser and Benner 2009). They are particularly high in surface waters (50%). Is this indicative of a different source of carbohydrates in surface waters? Given the similar yields (%DOC) among ocean basins it seems unlikely the high mol% glucose in the S. Pacific is due to greater diagenetic processing. The authors need to address the high mol% glucose values in the Discussion section.

The surface water of the MA area was characterized by a high abundance of Trichodesmium colonies (<200 $\mu$m to 2-5$\mu$m) size as shown by the underwater vision Profiler mounted on the CTD. The results showed that "fiber tricho-like Trichodesmium" (FTL Tricho) values ranged from 127 to 4125 Col m-3 in the MA whereas they were $\sim$ 0 in WGY area (Dupouy et al., 2018). Moreover, MODIS imagery acquired during the OUTPACE campaign revealed the presence of surface blooms northwest and

east of new Caledonia and near 20°S-172°W further indicating the presence of Trichodesmium in relation with the measured fixation rates (Rousset et al., 2018). Therefore the high abundance of glucose (∼ 50%) in surface MA waters may be due to presence of these species that potentially may release exopolysaccharides in the environment during their bloom or after their senescence. However, to the best of our knowledge the carbohydrate composition of these species is poorly known therefore, we do not have a solid evidence to support this statement. On the other hand, the high abundance of glucose in the surface WGY water ( ∼55%) is in agreement with the previous investigations (Sempéré et al. 2008) observed in the south Pacific gyre and may be due to presence of fresh organic material which is not taken up due to the limited nutrient availability. The above info is now provided in the revised MS (see page 15, lines 365-370).

I recommend the authors include a table with depth, chlorophyll, [DOC], [DCNS], DCNS %DOC, and mol% glucose data from all stations in the upper 200 m of the water column. It is not possible to derive quantitative values from the figures (2, 3).

We agree with this comment and a Table was now included in the revised version. Statistical analyses were made as well (Mann-Whitney test) to compare the MA and WGY areas. DCNS data are available at http://www.obs-vlfr.fr/proof/ftpfree/outpace/db/data/SUGARS/

The authors should take a look at the article by Shen et al. 2016 in L&O. This study shows the accumulation of carbohydrate-rich DOC during the summer in productive waters of the Gulf of Mexico. Nutrient limitation appears to play a role in carbohydrate-rich DOC accumulation.

We agree with this comment and we added this reference in the text (page 13, line 307) and support a part of our discussion using Shen et al (2016) results.

Specific comments: Methods: It is unclear how Dissolved Combined Neutral Sugar (DCNS) is calculated. The DCNS terminology implies that free sugars have been subtracted from total sugars. If so, this should be stated. If not, the DCNS terminology needs clarification because the term implies free sugars are not included. Clarification is important for comparison of values among studies using different terminologies.

We agree with this comment. In fact the desalination procedure described in page 6 (lines 128-143) does not allow the determination of dissolved free monosaccharides (i.e monosaccharide monomers present in samples with a MW ∼180 Da) because these compounds are lost/poorly recovered during the dialysis step (cut off of dialysis tubes 100-500 Da). This info is now included in the revised MS (page 6; lines 140-143).

Experiments in my lab with a STD glucose solution at $1\,\mu$M final concentration showed a recovery of 20-25 % (n=3) indicating that our approach is not well adapted to measure free monomers. Therefore the term DCNS used in this study corresponds indeed to combined monosaccharides found in polymers.

Line 31: change "also reflected" to "observed

DONE

Line 34: delete "high" and change "higher" to "longer"

DONE

Line 39: change "higher" to "longer"

DONE

Line 49: change to "life" to "productivity "

DONE

Line 72-73: The carbohydrate pool also includes oligosaccharides.

We agree with this comment and we added this information in the revised MS (see line 77).

Line 75: add reference McCarthy et al. (1996)

DONE

Line 78: change "that" to "they"

DONE

Line 79: delete "ultra"

DONE

Line 116-118: Are these values for multiple injections of a single sample or replicate samples? Report the average _ SD concentration for the reference standard.

DONE see lines 121-122 in the revised MS.

Line 154: provide a reference for the carbon conversion factor

We added the reference Kirchman, 1993 in the text (see page 7 line 166 in the revised MS) & reference section.

Line 165-166: include the range of mixed layer depths observed along the transect

We agree with comment and we added the mixed layer depth in the text (page 8, lines 182-184) as well in Fig. 3a in the revised MS along with the deep chlorophyll maximum.

Line 173: change "prokaryotic" to "bacterial" production (BP)

DONE (see page 8, line 187)

Line 182-183: reported range (55-78) is inconsistent with the highest value being 85, add the median value along with the range.

In the revised MS we provided the mean values of MA and WGY areas (Table 1).

Line 242-246: This sentence needs revision for clarity, e.g. 3H leucine concentrations. Duhamel et al. 2018 is not in the references. Total dissolved amino acids are known to contribute to semi-labile DOC and are measurable in mesopelagic waters of the Pacific.

The points raised by the reviewer were carefully addressed in the MS (see lines 279-282).

Line 194 and 196: add reference Kaiser and Benner 2009, values for the Pacific HOT station should be added for comparison

DONE. See revised MS page 9, line 216.

Line 211: Fig. 5 is presented before Fig. 4. Figure 4 is first presented in the Discussion and should therefore be presented as Fig. 5.

DONE

Line 251-252: Residence time (d) should be changed to Turnover time (d-1) because the calculation is based on microbial utilization (BCD) of DOC.

We computed the ratio DOCSL (mmol C m-2) to BCD (mmol C m-2 d- 1) which is in units of days (See Fig. 7 in the revised MS). It represents the time that would be necessary for the DOCSL pool to disappear completely due to its utilization by heterotrophic bacteria to satisfy their bacterial carbon demand, assuming no permanent renewal of this pool. The turnover rate (d-1) is simply the inverse of this ratio and it is not what we have decided to plot.

Fig. 1: The station symbols and numbers should be changed to colors that standout from the background.

The Figure has completely modified as the reviewer#2 suggested. It now contains a locator map (as reviewer#2 suggested) and we have changed the color of the sampled stations of the cruise. The legend was also changed accordingly.

Fig. 2: The legend states data are presented from 0-300 m depth, but the figure shows 0-200 m depth.

Corrected in the revised MS.

Fig. 3: The masking of data on figures 2B and 2C due to abnormal extrapolation of data is confusing and inappropriate. Rather than mask areas on the figures, these odv plots should be adjusted to properly extrapolate among profiles (i.e. not connected between profiles that are many kilometers apart).

We agree with the reviewer comment and we have adjusted the resolution to obtain adequate ODV figures (see Figures revised MS).

Fig. 4: Residence time (d) should be changed to Turnover time (d-1).

See reply above.

Fig. 5: The relative abundance of dissolved monosaccharides should be referred to as the mole percentage (mol %).

DONE, also corrected in the Fig. legend

Additional editing for grammar would improve the text.

DONE

---

## Author Comment (AC2) · 1 Oct 2018

The authors measured [DOC], [DCNS], and BP across a gradient of oligotrophic conditions in the South Pacific subtropical gyre. They calculate a variety of parameters relating to the role of bacteria in DOC cycling and carbon export in the region, including BCD and DOC residence times. The results and discussion of this manuscript need to better reflect the data presented in the figures – there are some notable disconnects, such as BCD, where the results in the figures are neither presented in the Results section nor is the significance of those results expounded on in the discussion. I've noted the instances that most stand out to me here:

[Figure]

1) The Results section needs to include BP, BCD, and PP (the latter is perhaps from another study, but it could still be summarized). My first thought on looking at Figure 4, in context of the authors' discussion of accumulating DOCsl, was that it seemed like BCD and PP might be reversed. While I quickly realized that's not the case, as the manuscript stands, there is no way to double-check the figure against numerical values.

We agree with this comment and in the revised MS we expanded our discussion on BCD and GPP (page 8, lines 192-197). The figure has changed accordingly (now Fig. 2), however the discussion on BCD and GPP are not the main points of this paper and the reviewer is invited to check on Van Wambeke et al. (2018).

2) The discussion section needs to be expanded, as the context and significance of the results are largely glossed over. In particular, the authors neglect to discuss the incongruity of BCDs that so largely exceed PP, yet they note a residence time of DOCsl of up to several months and a glucose-heavy DCNS pool that implies highly workedover DOM. So then what could possibly be supporting that high BCD in the gyre? Of course all of these measurements/estimations have uncertainties and assumptions, but those caveats need to be presented and/or the authors need to explain what they hypothesize is driving such extreme heterotrophy in the system.

We fully agree with this comment and in revised MS all of these points were carefully addressed:

- We explained how the DOC excess was calculated and this we assumed that corresponds to DOC semi-labile (page 11 lines 259-267). - We compare the MA and WGY areas using statistical tools (Man-Whitney tests) and the results are given in Table 1 - We expanded our discussion on residence time calculation indicating the uncertainties and assumption of our approach (lines 290-300 in the revised MS).

I recommend the authors conduct one more proofread for grammar – by and large the manuscript is well written, but there are many instances of missing words such as

prepositions.

Title: Would be more accurate to refer to DOC/DCNS rather than to labile DOM, as in fact the authors largely discuss presumed semi-labile DOM and make no direct measurements of DOM lability.

We agree with this comment and we indicated semi-labile DOM in the title.

Abstract: I would find it useful to have a brief summary of methods in the abstract (i.e., that residence times were estimated by comparing stocks rather than by experimental approaches).

DONE, see lines 32-36, page 2 in the revised MS.

Line 59: The abbreviation of gyre alone is very unnecessary. It saves 2 letters but makes the text significantly less reader-friendly.

We agree with this comment and we deleted this abbreviation from the whole text.

Line 67: I don't believe Goldberg et al. measured the percent of labile DOC constituted by carbohydrates, and further, the composition of most truly labile DOC isn't well characterized (e.g., reviewed by Carlson and Hansell 2015, cited in the manuscript).

In this sentence we simply indicated that the semi-labile of DOC is mainly represented by carbohydrates. There are several references that support this statement (Benner et al., 1992; Aluwihare et al., 1997; Skoog and Benner, 1997; Benner 2002; Aluwihare et al., 2005; Repeta and Aluwihare, 2006) including the references that we provided in text. Indeed Golberg et al. (2011) did not measure the % of labile DOC but their study showed a systematic removal of DCNS within DOM across the ocean basins pointing to semi-labile nature of carbohydrates. Note also that semi-labile DOC contains also carbohydrates that do not belong to the category of DCNS such as those measured in this study and these carbohydrates exist as unhydrolyzed polymers and methylated carbohydrates as shown by NMR (Panagiotopoulos et al., 2007).

Panagiotopoulos et al. 2007. Identification of methyl and deoxy sugars in marine high molecular weight dissolved organic matter (HMWDOM). Org. Geochem. 38, 884-896.

Line 89: This sounds like a hypothesis, but there is no discussion of N2 fixation in the results. Even if characterizing the N2 fixation gradient was the overall cruise goal, it should be removed from this manuscript unless the authors were to actually compare their results with N2 fixation via correlations or similar.

We agree with this comment and we deleted the word nitrogen fixation.

Results: A table with results in numerical format needs to be included. I was interested in seeing real numbers to judge for myself if the 4% difference in DOC concentrations between regions was consistent enough to be significant. It's much more difficult for others to use the work as a reference in the future if they are limited to estimating values from averages in figures. Finally, this is bad practice for data availability purposes. Would be fine to put it in supplementary material as long as this is clearly referenced in the text.

DONE, a Table with average volumetric values and ranges of DOC, DCNS-C, DCNS-C/DOC and DCGlc-c/DCNS-C was added in the revised MS. Statistics were performed as well to compare distributions in MA and WGY areas.

Line 252: I would like to see more discussion of the caveats of this estimation of residence time. E.g., the semi-labile pool is heterogeneous and composed of compounds that will be more or less biologically available, while BCD is derived from a measurement of BP that lasted 1-2 hours and therefore is likely based on the use of the most labile compounds available at that time. This may still be a useful metric for comparing between the two regions but it needs to be presented less as a clear-cut value.

We agree with the referee comment, as BP tracks labile to ultra-labile DOC substrates whereas DOC SL includes a pool turning over on time scales of weeks to months. In fact the DOCSL corresponds to the excess DOC (DOCEX) the latter calculated as the

difference between an average deep DOC value (39.6 ± 1.4 $\mu$MC, n = 36) from the bulk surface DOC pool. This info is now given in the revised MS (lines 246-249). Additional information is given in lines 263-297.

Regarding the estimation of the residence time we explained the assumptions that we made and we also included data of a biodegradation experiment (3 experiments 10 days each) performed also during the OUTPACE cruise. The results of this experiment (Van Wambeke et al., 2018; Table 5) showed that labile DOC represented only 2.5 to 5% of the DOC pool. This info is now included in the MS (see lines 290-300).

Lines 282-284: This sentence is confusing, please rephrase.

DONE, see page 14, lines 331-334 in the revised MS.

Line 303: How is this calculated? Is it DCNS divided by BCD, as for DOCsl above? That seems to make a lot of assumptions if so.

We agree with this comment. In the initial submitted MS we estimated DCNS residence time by dividing DCNS stock (integrated 0-200m) with BCD. We realized that this calculation was not corrected because (a) BCD was calculated after integration 0-euphotic zone and (2) only a portion of BCD is used for DCNS hydrolysis and subsequent bacterial uptake. In the revised MS (although some assumptions were made) we calculated DOC and DOCEX and DCNS-C stocks after integration 0-euphotic and we applied a percentage of 11% that corresponds to the part of BCD employed for glucose bacterial uptake after polysaccharide hydrolysis according to (Piontek et al., 2011). This info is now included in the revised MS (see page 15 lines 352-357). The results showed that a higher residence time of DCNS in the WGY (Tr = 91 ± 41 days, n = 3) than the MA area (Tr = 31 ± 10 days, n=8).

Line 306: The hypothesized role of carbohydrates in export to depth is not coherent with the statement three lines above that these compounds have residence times of 3 or 8 days, especially in a stratified system such as the gyre. Please expand on

this statement to explain this speculation better. (It's obviously quite possible this is happening, as DCNS are present at depth – perhaps this indicates an issue with the DCNS residence time calculation, as above?)

See previous comment on DCNS residence time calculation. The text is corrected accordingly (see page 15, 16, lines 361-364).

Line 324: I don't follow this sentence; please rephrase.

Done. See page 16, lines 384-389.

Line 541: Specify who/what is CLS.

The Figure was modified according to reviewer suggestions and its present legend does not contain this term.

Line 557: Carbon stock should not be d-1 here.

We agree with this comment and we corrected it in the revised MS.

Figures 1-3: The longitude should be marked consistently between Fig. 1 and Figs. 2-3.

DONE

Figure 1: A locator map would be appreciated for those reading this as a stand-alone paper and not as part of the broader cruise special issue.

DONE. Fig. legend was changed accordingly.

For Figs. 2 and 3, have you tried plotting panels B and C on the same axis? The ability to visually assess DCGlc as a proportion of DCNS might be worth the loss in resolution in panel C. (This is only a suggestion, please take it as such.)

In the revised MS we added this info in Table 1 (average values) and the reviewer can see the differences among MA and WGY areas.

Figure 5: It would help the reader to label the depths on each panel.

DONE.

---

## Author Response (AR2)

**Associate Editor Decision: Publish subject to minor revisions (review by editor)** (13 Nov 2018) by
Sophie Bonnet
Comments to the Author:
Dear authors,
Your manuscript has been sent to the original reviewers, who find that the manuscript has greatly improved. However one of them still recommends minor revisions (see below) before your manuscript will be accepted.
Best regards,
Sophie Bonnet

This revision is much improved – the more detailed descriptions of calculations and associated caveats are appreciated. I have a few comments on minor points that could use further clarification.

You might add a supplemental table with the actual statistical results, PDF document would be fine, just to make that info available for those who are interested. Likewise, will the actual data be archived somewhere? If so, you should point the readers to that repository; if not, you might consider putting the results in a supplemental file (i.e., concentrations at each depth at each station, in much more detail than Table 1). A major advantage of this work is that it contributes to our knowledge of an under-sampled region of the ocean, so those results will certainly be of use to the broader community.

Full DOC and DCNS data are available at the following link http://www.obs-vlfr.fr/proof/ftpfree/outpace/db/ and are in open access. This info is given in the revised MS in Table 1 legend.

Line 26: Please say 18degS instead of -18degN.

DONE

Line 29: Are you saying there's no difference in DOC concentrations at specific depths? Because at first glance this statement reads as inconsistent with the differences in integrated stocks reported just below. You do need to be careful throughout the manuscript to be clear when you're discussing concentrations (no differences between regions) vs. integrated stocks (significant differences between regions). E.g., Table 1 vs. Figure 6 – the distinction is clear with attention to detail, but I think it'd be worthwhile to make sure your readers have minimal opportunities to misinterpret. The body of the text is generally fine, it's more in the figures, abstract, conclusions, etc.

We agree with this comment and we re-phrased the sentence (lines 31-35 in the revised ms). In fact, in lines 29-31 we gave mean values of DCNS and DOC which were about similar in both areas whereas euphotic zone integrated values of same parameters changed considerably among the MA and WGY zone.

Line 125: This link doesn't work for me (it may be temporary, not sure). Might be less problematic to replace with a citation to something from Hansell's group talking about the ref waters.

Done. We corrected the link

Line 300: This is confusing and needs to be rephrased – you say you're calculating the residence time of DOCsl as DOCex, but then you talk about how the calculation overestimates the residence time of ultra-labile DOC. Are you trying to say that the residence time is over-estimated *because* it's calculated using uptake of ultra-labile DOC?

We agree with comment. Indeed the $DOC_{EX}$ is representative of $DOC_{SL}$ but at the opposite BCD is not accurately estimated for $DOC_{SL}$ because the leucine technique (incubation of 2 h) used for its determination tracks only the utilization of ultralabile to labile components. As such the residence time time of semi-labile DOC calculated as the ratio of DOCEX to BCD is overestimated. This point was better explained in the revised MS (see lines 292-306).

Line 353: biogeochemical rather than biochemical?

Changed according to the reviewer suggestion (see line 357 in the revised MS).

Line 365: This raises the question in my mind of whether the MA area experiences higher turnover or seasonal mixing than the gyre, to the extent that it would compensate for the shorter residence time of DCNSs in that region, such that the export might not be that different between the two regions, just controlled by different mechanisms? I am definitely not insisting that you go down that path, but it could be interesting.

Yes it would be an interesting perspective to link results of residence time to organic matter export or sequestration but with the current data in hand it is quite hard to explore further the reviewer's suggestion. In fact additional data including carbohydrate values in sediments traps, seasonal variability among the two areas will definitively help to go through this path.

Line 372-373: Can you cite something for the statement that exopolysaccharides would be hydrolyzed (suggesting the production of nitrogen-requiring exoenzymes) but then not taken up? Or if speculative, please be more explicit that that is the case. Or are you saying that glucose is a by-product of other exohydrolytic activity that isn't used and so accumulates? If the latter, this sentence needs rewording – took me a while to get to that possible meaning.

Yes this is speculative. We agree that if carbohydrate hydrolysis exists, some N should be available for the synthesis of ectoenzymes hydrolyzing it. (see revised MS lines 376-377).

Line 385: missing an adjective (higher/greater)

Indicated higher in the revised MS.

Lines 395-396: I would remove this sentence – it's not wrong, but it's not a meaningful conclusion from this particular study.

DONE

Figure 4: The relatively high deep-water DCNS concentrations at ~175W are intriguing – is there a water mass difference that could be contributing? Quite a lot higher than the values you cite in Line 218.

Yes, in the SD11 station DCNS values were around 2 µMC, we do not know why but all over the OUTPACE cruise mesopelagic DCNS ranged 0.3- to 2.4 µMC (line 216 in the MS).

Figure 6: Perhaps take a look at an axis break between the DOCex and DCNS-C values, to make DCNS-C easier to see, if differences or lack thereof between regions is the point? Unless the point of the figure is to emphasize that DCNS-C is a very small percentage of the overall DOC stock, in which case it works well for that purpose.

We agree with this comment and Fig. 6 was changed accordingly.

Throughout: Mann-Whitney is written as Man-Whitney in several places.

DONE

(see lines 627, 646, 649, 654)